# Review of Functional Aspects of Nanocellulose-Based Pickering Emulsifier for Non-Toxic Application and Its Colloid Stabilization Mechanism

**DOI:** 10.3390/molecules27217170

**Published:** 2022-10-23

**Authors:** Shao Hui Teo, Ching Yern Chee, Mochamad Zakki Fahmi, Satya Candra Wibawa Sakti, Hwei Voon Lee

**Affiliations:** 1Nanotechnology & Catalysis Research Center (NANOCAT), Institute for Advanced Studies, Universiti Malaya, Kuala Lumpur 50603, Malaysia; 2Department of Chemical Engineering, Faculty of Engineering, University of Malaya, Kuala Lumpur 50603, Malaysia; 3Department of Chemistry, Faculty of Science and Technology, Universitas Airlangga, Campus C, Mulyorejo, Surabaya 60115, Indonesia

**Keywords:** nanocellulose, Pickering emulsion, non-toxic application, stabilization mechanism, nanocellulose-stabilized Pickering emulsion, toxicity profile

## Abstract

In the past few years, the research on particle-stabilized emulsion (Pickering emulsion) has mainly focused on the usage of inorganic particles with well-defined shapes, narrow size distributions, and chemical tunability of the surfaces such as silica, alumina, and clay. However, the presence of incompatibility of some inorganic particles that are non-safe to humans and the ecosystem and their poor sustainability has led to a shift towards the development of materials of biological origin. For this reason, nano-dimensional cellulose (nanocellulose) derived from natural plants is suitable for use as a Pickering material for liquid interface stabilization for various non-toxic product formulations (e.g., the food and beverage, cosmetic, personal care, hygiene, pharmaceutical, and biomedical fields). However, the current understanding of nanocellulose-stabilized Pickering emulsion still lacks consistency in terms of the structural, self-assembly, and physio-chemical properties of nanocellulose towards the stabilization between liquid and oil interfaces. Thus, this review aims to provide a comprehensive study of the behavior of nanocellulose-based particles and their ability as a Pickering functionality to stabilize emulsion droplets. Extensive discussion on the characteristics of nanocelluloses, morphology, and preparation methods that can potentially be applied as Pickering emulsifiers in a different range of emulsions is provided. Nanocellulose’s surface modification for the purpose of altering its characteristics and provoking multifunctional roles for high-grade non-toxic applications is discussed. Subsequently, the water–oil stabilization mechanism and the criteria for effective emulsion stabilization are summarized in this review. Lastly, we discuss the toxicity profile and risk assessment guidelines for the whole life cycle of nanocellulose from the fresh feedstock to the end-life of the product.

## 1. Introduction

There has been a clear shift in the non-toxic application of lifestyles in developed nations that has led to concerns regarding the sustainability, renewability, eco-friendliness, health, and safety for humans and the ecosystem of products from the food and beverage, cosmetics, personal care, hygiene, pharmaceutical, and biomedical fields [1]. COVID-19 resulted in a high awareness among consumers regarding the quality of products that are used and consumed, especially for people post-COVID-19. Thus, consumers are more inclined to buy “green” grocery lists instead of fossil-based or petro-based products.

This has resulted in a growing market for various natural- or bio-based processing ingredients, such as bio-additives with multi-functions (emulsifier, thickener, rheology modifier, active agent carrier, fat controller, etc.), that are applied during the formulation of liquid-, gel-, and cream-based products for all types of non-toxic applications. Most products’ formulations are in the form of an emulsion that involves both immiscible aqueous and non-polar oil phases. Thus, the stabilization of surfaces and interfaces in the emulsion is a key issue for a wide variety of product development.

Generally, emulsion stabilization can be achieved by the addition of amphiphilic surfactants to reduce the interfacial tension of liquid–liquid interfaces. However, an alternative technology is now well established, in which surfactant-free dispersions can be stabilized by dispersed solid particles to form Pickering emulsions. The interface of Pickering particles must show a high affinity to oil–water phases, which form colloidal-sized particles that are anchored at both interfaces [2]. Many types of inorganic particles fulfil the partial wetting condition for most common oils, such as silica, calcium carbonate, clays (montmorillonite), carbon-based particles, latex, magnetic particles, and silver and gold nanoparticles, which have mostly been used as emulsifiers for the development of Pickering emulsions [3,4]. Research efforts are currently focused on the development of bio-based Pickering particles from renewable resources, which act as an alternative for petro-based polymers and non-biological solid particles, especially in the non-toxic fields (e.g., the food and beverage, cosmetic, personal care, hygiene, pharmaceutical, and biomedical fields) [5,6].

Natural fibers from lignocellulosic biomass constitute the most abundant renewable polymer resource (cellulose) available today, with low toxicity and high biocompatibility. Therefore, solid cellulosic particles with their low carbon footprint and low density have the potential to act as an emulsifier by chemically tuning the surface of cellulose to meet certain product formulations and specifications [6].

Thus, the objective of this review is to focus on bio-based Pickering emulsifiers from nano-emerged cellulose natural fibers (nanocellulose) for different types of safe applications. An overview of lignocellulose biomass and the characteristics of cellulose and different types of nanocelluloses based on the raw biomass feed, morphology and sizes, and preparation methods that can potentially be applied as a Pickering emulsifier is provided. In addition, studies of the use of nanocellulose-based Pickering emulsions (different water–oil systems) for various non-toxic applications are discussed. The hydrophilic nature of nanocellulose limits its water–oil dispersion and formulation. Thus, various types of nanocellulose surface modification with non-toxic organic moieties and functionalization with hydrophobic active compounds (e.g., drugs, antioxidants, hormones, enzymes, and vitamins) are highlighted. The aspects of an effective Pickering emulsifier, the stabilization mechanism of CNC- and CNF-stabilized emulsion systems, and the factors affecting CNC- and CNF-based Pickering emulsion are summarized in this review. Lastly, we discuss the toxicity profile and risk assessment guidelines for the whole life cycle of nanocellulose from the fresh feedstock to the end-life of the product.

### 1.1. Lignocellulosic Biomass

Lignocellulosic biomass is one of the most plentiful raw materials available on Earth that is not utilized efficiently. It can be obtained through many sources of dried plant matter such as softwood, hardwood, agricultural resides/wastes, and grasses [7]. In Malaysia, the oil palm industry contributes more than 90% of the country’s total lignocellulosic biomass [8]. The three main compositions of lignocellulosic biomass are lignin, hemicellulose, and cellulose, whose weight percentage differs according to the age, culture conditions, and harvesting seasons [7]. Table 1 shows the chemical composition of several lignocellulosic biomass based on the dried weight. However, in general, lignocellulosic biomass is made up of 40–50% cellulose, 20–30% hemicelluloses, 10–25% lignin, and small amounts of proteins, pectin, and extractives [9]. These contents bind with each other via strong intermolecular forces and can be found in the vicinity of the plant cell walls.

Lignin, a molecule with a complicated structure, is made up of phenolic monomers that cross-link with each other via ether and C-C covalent bonds. The three common monomers are coniferyl alcohol, paracoumaryl alcohol, and sinapyl alcohol [27]. Due to the complexity and rigidity of its molecular structure, it contributes to the low permeability of plant cells, low susceptibility towards microbial attacks, and high mechanical properties [28].

Hemicellulose is a hetero-polysaccharide consisting of monosaccharides such as pentoses (xylose and arabinose), hexoses (glucose, mannose, and galactose), desoxyhexoses (rhamnose and fucose), and uronic acids [29]. It is an amorphous branched polymer, containing acetyl groups along the chain [30]. It cross-links with lignin and forms microfibrils with cellulose, resulting in high stability of the biomass [31]. The branched polymer structure and low degree of polymerization results in a low crystallinity index for hemicellulose, which can be easily removed from the complex biomass under mild conditions [32].

Cellulose is a long and stable polysaccharide comprising more than 10,000 subunits of glucose that are linked through β-1,4 glycosidic bonds [29]. It is a linear polymer, with hydroxyl groups positioned along the polymer chain, which can form inter- and intra-molecular hydrogen bonds between the fibers. Due to the presence of all types of chemical bonds, the cellulose chains interlink with one another to form microfibrils in crystalline and amorphous forms [9]. The amorphous regions of the cellulose fibrils can be easily hydrolyzed comparatively to the crystalline region as they are more accessible during chemical attacks.

For lignocellulosic biomass to be converted into useful or intermediate products, it must overcome lignocellulosic biomass recalcitrance. Biomass recalcitrance is considered the natural resistance of plant cell walls to microbial and enzymatic deconstruction [30]. It is influenced by the physiochemical properties of the plant cell walls, such as the size of the particle, volume of the pore size, specific surface area, crystallinity, degree of polymerization, and the composition of lignin, hemicellulose, and the acetyl group present in the cell walls [33]. Thus, pre-treatment processes are usually performed to overcome these problems for conversion into useful products and intermediates, such as alcohols (ethanol, sorbitol, ethylene glycol, etc.), acids (levulinic acid, lactic acid, formic acid, acetic acid, glutamic acid), furfural (5-(hydroxymethyl furfural), 5-(chloromethyl)furfural), and nanomaterial (nanocellulose) [34,35]. Figure 1shows several platform chemicals that can be produced from cellulose.

To access the cellulose from the complex biomass, the recalcitrant structure of the plant cell wall needs to be disrupted via the pre-treatment process, in which destruction of the lignin sheath, decomposition of hemicellulose, and reduction of the cellulose crystallinity and degree of polymerization occurs. Countless pre-treatment processes are applied for different types of biomass, ranging from simple physical-mechanical processes (e.g., microwave irradiation, ultrasound sonication, ball milling) to chemical treatments (e.g., alkali or acid hydrolysis, chlorite oxidation, hydrothermal process) or biological treatment (e.g., enzymatic hydrolysis) [7,32,36].

As cellulose is rich in carbon and modifiable hydroxyl groups, it can be considered a natural source of raw material for the production of many valuable carbon-based products that are eco-friendly and safe for human usage [37]. Other than the application of cellulose for bio-based chemical platforms and biofuels (liquid form), solid cellulose ester derivatives and ether derivatives work actively for various types of non-toxic applications, such as as thickeners, emulsifiers, stabilizers, drug carrier suspending agents, sustained-release agents, and film-forming and binding agents for food, beauty, and personal care products; health care and hygiene; and the pharmaceutical industry [38,39,40].

The rapid development of nanotechnology has resulted in a high emerging demand for bio-based nanoparticles in various non-toxic fields due to their extraordinary characteristics (e.g., lighter in weight, stronger, durable, more reactive, etc.) at a nanoscale. Thus, nano-dimensional cellulose (nanocellulose) has attracted much attention from researchers in the past few decades as a novel, non-toxic, environmentally friendly, and advanced biomaterial that is applicable in various fields [41]. Nanocellulose has a high self-assembling ability at liquid–liquid interfaces, high aspect ratios in the nano-dimension in width or length, large surface area, tunable functional groups for surface modification and grafting that allow further control to attain supra-structures, and highly hierarchical assembly. The superior properties of cellulose on a nano-scale result in this bio-material having potential as a thickener, emulsifier, stabilizer, protective colloid, suspending agent, gelling agent, flow control agent, and film-forming and binding agent for food, beauty, and personal care products; health care and hygiene; and the pharmaceutical industry [39].

### 1.2. Nanocellulose

According to the Technical Association of the Pulp and Paper Industry (TAPPI), the terms of cellulose nanomaterials were established and defined under TAPPI WI 3021, where cellulose nanomaterial or nanocellulose is defined as a material that is nanoscale in either its external dimension or internal structure. Nanocellulose can be classified into nano-structure material or nanofiber. Cellulose microcrystalline and microfibril are examples of nano-structure material while cellulose nanocrystal (CNC), cellulose nanofibril (CNF), and bacterial cellulose (BC) are examples of nanofiber [41]. The three mentioned nanofibers have similar chemical compositions but have distinct physical properties such as the morphology, crystallinity, and flexibility because of the different extraction methods and sources [37].

Cellulose nanocrystal (CNC), also known as cellulose nanocrystalline, has a morphology of 2–20 nm in diameter, 100 nm to several micrometers in length, and 54–88% crystallinity [37,42,43]. It exhibits a high level of crystallinity, surface area, strength, and hardness [43]. It can be obtained through acid hydrolysis (sulfuric acid, hydrochloric acid, or phosphoric acid) as the hydronium ions, H_3_O^+^ ions, penetrate the amorphous region of the fibrils through hydrolytic cleavage of the β-1,4 glycosidic bonds [44]. The use of highly concentrated acids in this process is not ideal as they are not environmentally friendly and difficult to recycle. Hence, a novel green method has been studied for the preparation of CNC from lignocellulosic biomass. A few examples are solution plasma technology, cooking with active oxygen and solid alkali (CAOSA), and deep eutectic solvent (DES)-based treatment [31].

Cellulose nanofibril (CNF), also known as nanofibrillated cellulose, exists as long and flexible nanocellulose, 20–100 nm in diameter and >10 µm in length, with a lower crystallinity compared to CNC [41]. It is produced from the splitting of cellulose microfibrils from the mechanical process, thus containing both the amorphous and crystalline parts of the fibrils [45,46]. This results in a higher aspect ratio and unique rheological properties but lower crystallinity and poor mechanical properties [47]. A variety of mechanical methods are used to produce CNF. A few examples are the high-pressure homogenizer, microfluidizer, ultrasonication, and cryocrushing [48,49]. These methods require high energy consumption, which makes them infeasible economically. Therefore, research has been carried out on the use of chemical and biological pre-treatments such as enzymatic hydrolysis and TEMPO-mediated oxidation prior to the mechanical process to reduce the amount of energy used [49].

Bacteria cellulose (BC), also known as microbial nanocellulose, comprises pure, ultrafine, and ribbon-shaped nanofibers with a diameter of 30–50 nm and length in the micrometer range [41]. BC is synthesized from a bottom-up approach using bacteria such as *Gluconacetobacter xylinius* to build up the cellulose fibrils from low-molecular-weight sugars for several days, which can be carried out using three different culture methods (static, agitated, and bioreactor cultures) [50]. As for its physiochemical properties, it has a higher crystallinity and purity, high water absorption capacity, great flexibility, and higher amount of hydroxyl groups than the other two mentioned nanocellulose [51,52]. Even though it may seem that BNC is more superior to CNC and CNF, it is not efficient to produce it on a large-scale basis because of the low productivity and the high cost of the media required for the cellulose to form [50].

The three types of nanocellulose have different physical characteristics due to their source of raw material and preparation process. However, there are other types of nanocellulose that can be synthesized, which have been studied and reported less. Amorphous nanocellulose is one type of nanocellulose with a spherical or elliptical shape, with a diameter ranging from 50 to 200 nm [47,53]. It can be formed from acid hydrolysis followed by ultrasonic disintegration of cellulose [54]. Owing to its amorphous structure, it possesses unique properties such as high accessibility, a high degree of functionalization, and improved sorption ability but poor thermal stability and mechanical properties [53]. Another example is the cellulose nanoyarn or electrospun cellulose nanofibers, which is obtained through the electrospinning of cellulose solution or its derivatives [41]. This process produces relatively bigger-sized nanocellulose with a diameter of 100–1000 nm and a length exceeding 10 µm. It is highly porous due to the mats of tangled long nanofibrils but has poor mechanical characteristics and thermal stability [47].

Cellulose nanoplatelets, one of the less studied nanocellulose types, is formed of entangled cellulose nanofibrils, which can be prepared by acidified oxidation under mild conditions [55]. The nanoplatelets have a thickness of 70–80 nm and the cellulose fibrils have a diameter of 2–3 nm. In addition, van de Ven and Sheikhi’s study reported a novel type of nanocellulose called hairy cellulose nanocrystalloid (HCNC), with a morphology of 5–10 nm in diameter and 100–200 nm in length [56]. It contains both crystalline and amorphous regions as it mainly has a rod-shaped dimension but with protruding polymer chains on both ends of the rod. HCNC possesses morphological characteristics from both CNC and CNF, resulting in unique surface properties, high mechanical properties, and colloidal stability. Table 2shows a summary of the characteristics and synthesis processes of the different types of nanocellulose for easier comparison while Figure 2shows the SEM images of amorphous nanocellulose, cellulose nanoplatelets, and electrospun cellulose nanofiber.

Researchers have been encouraged to implement the use of nanocellulose for non-toxic applications due to its extensive physical, chemical, and biological properties. Due to their nanosize, they possess unique characteristics, including high mechanical strength, high surface area, surface chemical reactivity, barrier properties, renewability, biodegradability, biocompatibility, and non-toxicity [58,59]. As nanocellulose is known as a bio-based polymeric material equipped with unique properties and biocompatibility and renewability, it can be used in a variety of ways in many eco-friendly fields, especially in the food, cosmetics, and pharmaceutical industries. In the food industry, it has been used as food additives and in food packaging materials [60]. In addition to this, its low cytotoxicity and high biocompatibility have attracted much attention in the biomedical field, where it has been employed in health care applications [46]. Due to its nano-size, it is also able to act as an effective drug carrier that can penetrate through skin pores either for medicine or cosmetic uses [61,62].

As nanocellulose bears many hydroxyl groups on its surface, it tends to aggregate in non-polar solvents, which limits its application and use. To overcome this problem, it must undergo surface modification to alter its hydrophilicity characteristic through chemical, physical, or even biological interactions [41]. There are numerous approaches to nanocellulose functionalization such esterification, etherification, silylation, grafting of polymers, and others [45,63,64]. Figure 3displays a few common surface modifications of nanocellulose via chemical, physical, and biological approaches.

The modification of nanocellulose results in a wide range of wettability in the water–oil phases while remaining insoluble in either phase. This results in the bio-based nanomaterials irreversibly adsorbing on the water–oil interfaces, and encapsulation of the droplets to form a dense adsorption layer that acts like a physical barrier to reduce agglomeration and coalescence between the droplets, resulting in the formation of colloid, namely Pickering emulsions. Thus, nanocellulose and its derivative materials can potentially act as renewable Pickering emulsifiers, which fulfil consumers’ demand for safe, non-toxic, eco-friendly, and green-label products.

### 1.3. Pickering Emulsion

An emulsion is a heterogeneous mixture of two or more immiscible liquids, with one of them being dispersed in the other. Some common examples of this are oil-in-water mixtures that are used daily in our lives such as sauces, creams, and vaccines. This system is highly regarded as being thermodynamically unstable due to the high surface energy between the liquids [65]. Small molecular emulsifiers or stabilizers such as chemical surfactants are traditionally used to stabilize an emulsion by reducing the interfacial tension. However, most surfactants are toxic to humans and the environment and carcinogenic, which may not be suitable for human consumption and daily usage. Thus, an alternative to the stabilization of the emulsion is necessary.

Pickering emulsion is another type of emulsion, in which solid particles are utilized for the stabilization of emulsions instead of surfactant molecules. This type of emulsion has been receiving increasing interest from researchers in various fields due to its favorable features such a lower toxicity, lower cost, and simpler recovery. There are several reasons why Pickering emulsion is favored more compared to the conventional surfactant-based emulsion as it provides more advantages for the emulsion. First, the solid particles are irreversibly adsorbed at the interface between the two immiscible liquids, forming a barrier that limits coalescence between the droplets. This results in a decrease in the high surface energy of the oil–water interface and increases the stability of the system [66,67]. In addition to this, the type of particles used can improve the characteristics of the system such as the conductivity, responsiveness, porosity, and others [68]. Moreover, food-grade solid particles can be used for in vivo emulsion applications for higher safety purposes. A few examples of these uses are drug delivery, controlled release, porous tissue scaffolding, and microencapsulation. Moreover, other non-medical uses are filtration membranes, catalytic facilitation, catalytic separation, and extraction [68].

Emulsions can be categorized into several types as shown in Figure 4. Water-in-oil (W/O) and oil-in-water (O/W) emulsions are the more common types of emulsions, which are stabilized by hydrophobic particles and hydrophilic particles, respectively. There is a much more complex emulsion system called double emulsion, in which droplets of the dispersed phase contain smaller droplets with different emulsions [69]. Water-in-oil-in-water (W/O/W) and oil-in-water-in-oil (O/W/O) emulsions are the major types of this emulsion, which are usually stabilized with a combination of two different surfactants or unmodified and modified particles.

The solid stabilizing particles are essentially much smaller than the emulsion droplets, where nanometric- or sub-micron-sized particles can stabilize droplets with diameter that is a few micrometers while micron-sized particles are able to stabilize droplets with a diameter that is a few millimeters [70]. There are many types of solid particles that are suitable for use in Pickering emulsions. The most common examples are inorganic particles such as metal oxides, clay, and silica; however, their use in food products that have direct contact with humans and animals is limited due to their potential health risks and food labeling issues [66,71]. Polymer-based particles such as polystyrene are also potential emulsifiers but are not suitable for similar reasons to inorganic particles [4]. Another example is protein-based particles from zein, whey protein isolate, soy, and gliadin. Although they are safe for human consumption, they are difficult to produce on a large scale because of challenges such as the cost-effectiveness and reliable processing operation [72]. Thus, naturally derived polysaccharides such as chitosan, starch, and cellulose have been used in Pickering emulsions, ensuring both food and product safety for consumers due to their abundance in the world and ease of processing [73].

### 1.4. Application of Nanocellulose-Based Pickering Emulsion

Plant-based and agro-industrial-derived nanocellulose have been studied extensively for applications in Pickering emulsion. Nanocellulose is generally considered to be hydrophilic, but its crystalline allomorph showed that it has amphiphilic properties, making it suitable as a Pickering emulsion particle. Nanocellulose has the ability to produce emulsions with high stability that can last for several months under various extreme conditions, which may be attributed to steric and electrostatic effects [74]. Generally, nanocellulose (e.g., CNF) with a high aspect ratio forms a percolating network that increases the cohesion and long-term stability of the emulsion [75]. Moreover, essential criteria such as the renewability, low toxicity, chemical stability, and biocompatibility allow the application of cellulose nanomaterials in emulsion stabilization [60]. These properties are highly sought after for a wide range of cream- and gel-based cosmetics, personal care and hygiene products, drug carriers in drug delivery systems, and bio-active agents for food and pharmaceutical applications, where there is direct contact between the products and humans or animals. Several reports have indicated that nanocellulose is suitable as a Pickering stabilizer for general use in different types of applications that suit human usage [76,77,78,79]. However, the present review focuses on studies of nanocellulose-stabilized Pickering emulsion and its potential usage in either food, cosmetic, or biomedical applications.

#### 1.4.1. Food Industry

The ability of nanocellulose to form a gel at low nanoparticle concentrations has been identified as a very interesting material to be used in food applications as a fat replacer [60]. Lignocellulosic biomass wastes such as lemon seeds [80], pistachio shells [81], banana peel [82], and ginkgo seed shells [83] have been used to prepare nanocellulose for food emulsion applications. These materials might contain phytochemicals, which have potential functional properties and provide beneficial effects in food products. On the other hand, materials other than lignocellulosic biomass wastes such as *Acetobacter xylinus* (bacteria) have also been used to prepare nanocellulose to study the stability efficiency in dodecane-in-water [84]. These studies proved that nanocellulose is suitable as aa Pickering stabilizer in food emulsions where environmental factors such as the pH, ionic strength, and temperature were highlighted to improve the stability of emulsions. Li et al. reported that a pH ranging from 3 to 10 and an ionic strength ranging from 20 to 200 mM affected the zeta potential of BC but had a negligible influence on the stability of the emulsion [84]. However, Dai et al. showed that a high pH (8–10) or high ionic strength (30–50 mM) enhanced the emulsion stability of CNC and CNF [80].

Nanocellulose is a bio-based polymer with potential in the food packaging industry, where it can be incorporated into the packaging material to improve its mechanical properties, thermal properties, gas barrier, and water vapor barrier properties in addition to forming active packaging material with controlled release and responsive packaging [85]. Zhang et al. reported the preparation of a poly(lactic acid)/CNC composite via the Pickering emulsion method and found that this approach improved the dispersion of CNC within the matrix of the complex when compared to the melt mixing and solvent casting method [86]. Some studies have integrated nanocellulose-stabilized essential oils into packaging material [87,88]. The nanocellulose-stabilized essential oils not only resulted in an improvement in the mechanical properties but also the biological activity (antioxidant, antimicrobial) as well. Sogut reported that whey protein isolate films with nanocellulose-stabilized bergamot oil had a better water resistance and mechanical properties and displayed a slower release rate and higher antimicrobial and antioxidant activity when compared to films directly mixed with bergamot oil [87]. Souza et al. showed that starch-based films containing nanocellulose-stabilized Ho wood essential oil displayed excellent mechanical behavior, strong chemical interaction, and a 69.0% influence on the water vapor permeability properties when compared to starch-based films containing nanocellulose-stabilized cardamom and cinnamon essential oil [88].

#### 1.4.2. Pharmaceutical/Medical Industry

In the biomedical/pharmaceutical fields, emulsions are used in many products such as creams, gels, vaccines, and others. Emulsions such as these are usually used for the encapsulation of active pharmaceutical ingredients (APIs) in the dispersed phase to prevent degradation and control the release rate of APIs within the body. These emulsions are usually stabilized by synthetic surfactants such as sodium lauryl sulfate, cetylpyridinium chloride, betaine, sorbian fatty acid ester, and many more [89]. These synthetic surfactants have high toxicity and low biodegradability, which makes natural stabilizers that have low toxicity and high biodegradability much more preferable. The encapsulation of bioactive compounds in oil/water emulsions by nanocellulose has been researched in several studies [90,91]. These studies reported the formation of stable emulsions with antibacterial or antimicrobial activities. Contrarily, a study reported by Winuprasith et al. focused on the encapsulation of vitamin D_3_ in oil by CNF, controlling its stability and bio-accessibility [92]. The application of NC-based Pickering emulsions can also be used as a template to form bio-based aerogels as a mechanically strong thermal super-insulating material [93].

#### 1.4.3. Cosmetic Industry

There are fewer studies of the use of nanocellulose-stabilized emulsion in the cosmetic field, however, compared to the food and biomedical fields. Products such as creams, tonics, nail conditioners, and others are a product of cosmetic emulsion, in which nanocellulose can be applied as Pickering stabilizers [61]. However, limited studies have been carried out on its use in cosmetic formulations even though it has been extensively studied as a suitable stabilizer in food applications. Despite its feasible use in cosmetic formulations, it can also be used for the formation of nanocomposites such as latex via Pickering emulsion. A study by Kedzior et al. utilized CNC and methyl cellulose as stabilizers in the formation of poly(methyl methacrylate) latex using the Pickering emulsion method [94]. They found that the Pickering emulsion is a greener method for incorporating CNC into polymer latex when compared to solvent-based polymerization. They also reported that this method can be used to formulate other cosmetic emulsion products such as toners, adhesives, and coatings. Table 3 summarizes the results of recent research articles on the use of nanocellulose as a Pickering emulsifier in food, cosmetic, and biomedical applications.

Despite nanocellulose-stabilized Pickering emulsion having high biocompatibility and assured safety usage that offers a wide range of possibilities in non-toxic applications, it is important to understand the interfacial stabilization between nanocellulose particles with a water–oil system. Compared with conventional emulsifier/surfactant, the nanocellulose-based Pickering emulsifier can be irreversibly adsorbed at the water–oil interface to form a stable film or colloid for different types of formulations and usages. In addition, comprehensive investigation of nanotechnology-based Pickering emulsifiers in the interface’s linkage, emulsion formation, and mechanism of action is still lacking. Thus, the present review also emphasizes the factors or criteria for better emulsion stability profiles, which are discussed in later sections.

## 2. Characteristics of Effective Pickering Emulsifiers

The formation of Pickering emulsion is different compared to surfactant-based emulsification as displayed in a schematic diagram in Figure 5. Surfactants are surface-active compounds that lower the interfacial tension between the two immiscible liquids and the energy required for the emulsion. They position themselves between the liquids, with the hydrophilic head facing the water and the hydrophobic tail facing the oil phase. Unlike surfactant-based emulsion, the solid particles required for Pickering emulsion can be wetted in both phases and irreversibly adsorbed at the interface of the two phases to form a steric or electrostatic barrier. This shows that the formation of surfactant-based emulsion and Pickering emulsion and the requirements for the surfactant and solid particles to be ideal emulsifiers are different.

Even though both surfactant-based emulsion and Pickering emulsion have different formations, they both experience emulsion instability or demulsification, where the separation of both immiscible phases can be observed [99]. Four types of emulsion instability can occur, namely, coalescence, Ostwald ripening, flocculation, and creaming/sedimentation as shown in Figure 6. Coalescence occurs when droplets of emulsion are in contact with each other and spontaneously merge to become a larger droplet. Ostwald ripening, on the other hand, is the diffusion of small droplets to larger droplets at a slower rate. These two instability types cause the average droplet size to increase as time progresses. Flocculation is the clumping of emulsion droplets due to an attractive force, forming flocs in the process. Creaming or sedimentation occurs when the droplets move towards the top of the emulsion because of buoyancy or deposit at the bottom of the emulsion because of centripetal force, respectively. Creaming usually occurs in an oil-in-water emulsion while sedimentation is typically observed in a water-in-oil emulsion. The average droplet size in these three types of demulsification does not increase unlike in coalescence and Ostwald ripening. Emulsions are bound to become unstable over time because they are thermodynamically unstable due to the interfacial surface tension between the two immiscible liquids. Therefore, the emulsion system will separate and the interfacial energy will decrease. The emulsifier plays an important role in achieving an emulsion that prolongs the stability and delays the separation of the two liquids. In this review, we will only focus on the characteristics of the solid particles used in Pickering emulsion.

Solid particles must possess certain properties in order for them to be a stabilizer of Pickering emulsion [65]. First, the particles must be able to be partially wetted by both the continuous and dispersed phases while at the same time possessing insolubility in either of the phases. Second, the surface charge of the particles must be sufficient for them to be adsorbed onto the interface between the two liquids but not too high that electrostatic repulsion for each another is induced. Third, particles need to be significantly smaller than the desired emulsion size. These properties determine the assembly and adsorption process at the interface of the liquids [100].

The irreversible adsorption of solid particles in Pickering emulsion can be attributed to the high detachment energy of the particles from the interface (ΔG). This energy can be calculated by the following equation [100]:ΔG = ±πR^2^γ (1−|cos θ|)^2^,(1)
where ΔG is the detachment energy of the particles, R is the radius of the particles at the oil–water interfaces, γ is the interfacial tension between the oil and water, and θ is the wetting contact angle, which will be explained later.

However, this equation is only applicable to particles that have perfect spheres. With an increase in studies on the type of particles used in Pickering emulsion, there are now various particles with different morphology and modification of the equation must be carried out to accommodate the different types of particles [100]. One example is shown as follows for the estimation of the detachment energy of rod-like particles:ΔG = ±lbγ (1−| cos θ|),(2)
where l is the length and b is the width of the rod-like particles. From Equation (1), a small nanoparticle with a radius < 1 nm is estimated to have the same detachment energy and thermal energy [101]. The desorption energy is approximately equal to 10 kBT, where kB is the Boltzmann constant (1.38 × 10^−23^ m^2^ kg s^−2^ K^−1^). This will result in an unstable emulsion system as the low detachment energy enables fast adsorption and desorption of the particle at the interface. Alternatively, for a particle to have a radius of ~10 nm, the desorption energy must increase to be several magnitudes larger than the thermal energy (up to ~104 kbT) [102]. This results in a strong attachment to the interface and irreversible adsorption onto the surface and a highly stable Pickering emulsion [103].

The adsorption rate of the particles at the interface should also play a role in the formation of Pickering emulsion. The adsorption rate should be faster than the coalescence rate of the droplets of the dispersed phase because coalescence may occur before particles can adsorb onto the interface [104]. This can be solved by introducing a high level of energy such as homogenization techniques to overcome the initial particle adsorption energy barrier [103].

### 2.1. Surface Wettability

As mentioned above, solid particles must be wettable in both the continuous and dispersed phases while at the same time being insoluble in either phase. The surface wettability of the particles is essential not only for the formation of the Pickering emulsion but also determines the type of emulsion that is formed.

According to the Bancroft rule, hydrophobic particles are more suitable for stabilizing water-in-oil emulsions while, on the other hand, hydrophilic particles are preferred for stabilizing oil-in-water emulsions. The rule also states that when mixing the same volume of oil and water, a long-term stable emulsion is more likely to formulate [67].

According to Finkle’s empirical rule, favorable surface wettability properties of the particle present in the bulk continuous phase will establish an effective mechanical barrier on the interface of the dispersed liquid [105]. Just like the Bancroft rule, it also states that the type of emulsion is heavily dependent on the relative particle wettability in both phases. The continuous phase of the Pickering emulsion will be determined by which phase the particles dispersed more in and not the ratio of the two liquids [106].

The relative particle wettability can be estimated through the three-phase contact angle (oil–particle–water), θ, which is equivalent to the hydrophilic–lipophilic balance for surfactants as it indicates the affinity of the particles for both water and oil [104]. The wetting contact angle in water can be calculated by Young’s equation assuming the shape of solid particles is a perfect sphere [107]. Young’s equation is as follows:(3)$$\cos\theta_{w}=(\frac{\gamma_{s-o}-\gamma_{s-w}}{\gamma_{o-w}}),$$
where γ_s−o_, γ_s−w,_ and γ_o−w_ represent the solid–oil, solid–water, and oil–water interfacial tensions, respectively. From the three-phase contact angle, we can determine the type of emulsion that will form. For single-layered particle emulsion, the contact angle of the particles at 15° < θ < 90° will preferentially form oil-in-water (O/W) emulsion while the contact angle at 90° < θ < 165° will preferentially form water-in-oil (W/O) emulsion [108]. Conversely, for multi-layered particle emulsion, the contact angle of the particles at 15° < θ < 129° will preferentially form oil-in-water (O/W) emulsion while the contact angle at 50.7° < θ < 165° will preferentially form water-in-oil (W/O) emulsion. On top of this, a contact angle at 30° < θ < 150° will result in irreversible adsorption of particles at the interface as the desorption energy is larger than the thermal energy of Brownian motion by several orders of magnitude [108].

However, many methods that have been employed to determine the contact angle more accurately instead of the estimation using Young’s equation. The captive drop method and gel trapping technique are methods that can be used to determine the wetting contact angles. Nowadays, with the aid of modern instruments such as optical microscopes and optical tensiometers, direct measurements of the wetting contact angle of the solid particles at the interface are much preferred to obtain a more accurate result.

### 2.2. Surface Charge

For the formation of a Pickering emulsion, the surface charge influences the adsorption of solid particles onto the interfaces between two phases and the colloidal properties of solid particles [65]. The net surface charge of a particle could induce an electrostatic attraction with the oppositely charged surfaces or electrostatic repulsion against the same charged surfaces [109]. Thus, the magnitude of the surface charge of the particles is crucial to forming a stable colloidal dispersion, and this can be determined by measuring its zeta potential. The zeta potential is a measure of the surface charge on nanoparticles in solution [110].

A particle suspension with a zeta potential value of more than ±30 mV will have high surface charge particles that experience electrostatic repulsion with one another and prevent agglomeration. However, solid particles with an extremely high zeta potential might undergo repulsion with each other and interrupt the adsorption process at the liquid–liquid interfaces. On the other hand, particle suspensions with a zeta potential value ranging from −30 mV < Zp < 30 mV are considered to be low, where the particles will most likely be dominated by the van der Waals forces and result in a cluster of dispersed particles, causing agglomeration of the particles to occur before the adsorption of particles. However, the cluster of dispersed particles may improve the stability of the emulsion by forming a network within the continuous phase.

The surface charge of a solid particle can be altered through substitution by the adsorption or grafting of charged groups onto the surface of the particles. The degree of substitution or the surface charge density can be determined by conductometric titration, which is also known as conductometry. Conductometry has several advantages as it is a fast and simple process that measures the conductivity (conductance) of the charged group in a solution without the need for expensive equipment [111]. This can be achieved by a titration process, where one ion will be replaced with another ion with a different conductance. The conductance changes linearly throughout the titration process until the replacement is complete, at which point the titration line will have a different slope [112].

### 2.3. Dimension of Solid Particles

The dimension of the solid particles is also important in the formation of Pickering emulsion as it controls the characteristics of the emulsion such as the emulsion droplet size, the stability of the emulsion, the stabilization mechanism, and the concentration of particles required for a stable emulsion. From Equation (1), the increase in the size of the particles (radius in this equation) will also increase the detachment energy of the particles from the interface, resulting in a high-stability Pickering emulsion. However, this is not true for every Pickering emulsion formed. Smaller particles are able to form more stable or at least stable emulsions as they are better at preventing coalescence of the emulsion droplets [113]. On the other hand, larger particles may form highly stable emulsions, but particles beyond a critical dimension will lead to an unstable emulsion as Brownian motion is superior and prevents the positioning of the particles between the interfaces. Their low diffusivity onto the interface results in slower adsorption kinetics and inefficient packing [113].

The dimension of particles affects the size of the droplets formed during emulsification, by which the emulsion droplet size increases proportionally to the dimension of the particles used for stabilization. Their relationship can be seen from the equation as follows:(4)$$r_{e}=\frac{4\phi_{d}r_{p}}{\phi_{p}},$$
where r_e_ and r_p_ are the radius of the emulsion droplets and Pickering particles, respectively, whereas ϕ_w_ and ϕ_p_ are the volume fraction of the dispersed phase and particles, respectively.

However, it was found that this equation becomes invalid after some time, which could be due to the change in the contact angle with the particle, which directly affects the number of particles occupying the liquid–liquid interface [65]. Despite this, a decrease in the size of the emulsion droplet with the particle size becomes a criterion, whereby the particle size should be at least an order of magnitude smaller than the desired emulsion droplet [114]. Thus, it can be considered that there is general agreement between the size of the emulsion droplets and the stability of the emulsion, where the desired Pickering emulsion can be obtained from a suitable solid particle size [65].

The dimension of the particles also has an impact on the concentration of the particles required for the stabilization process as the radius of the particle is proportional to the concentration [115]. Their relationship can be observed in the equation as follows:M_p_ = (16/3) πr_p_ρ_p_r_e_^2^n_e_,(5)where r_p_ is the average diameter of the particles, r_e_ is the radius of the emulsion droplet, n_e_ represents the number of droplets in the emulsion, m_p_ is the mass of the particles (concentration), and ρ_p_ represents the density of the particles.

## 3. Mechanism of the CNC/CNF-Stabilized Oil–Water System

The stabilization of Pickering emulsion by solid particles can be classified into three mechanisms: (A) stabilization by interfacial film/envelopes of particles surrounding droplets of the dispersed phase, (B) encapsulation of the droplets in a three-dimensional network, and (C) stabilization by the depletion effect of particles as shown in Figure 7 [72,114]. Stabilization by envelopes of particles consists of colloidal particles being adsorbed at the interface of the two immiscible phases, forming a single layer of several layers [116]. The colloidal particles then experience electrostatic repulsion or steric hindrance, which prevents the agglomeration of the dispersed phase depending on the type of solid particle used. On the other hand, the encapsulation of droplets in a three-dimensional network refers to the formation of a three-dimensional network between the droplets, which inhibits their movement [114]. Lastly, stabilization by the depletion effect is a comparatively less-studied mechanism of Pickering emulsion. This mechanism increases the attraction between the particles by adding a non-absorbing polymer, which forms a 3D particle network that inhibits the movement of emulsion droplets [71].

Lu et al. studied the interfacial percolation of CNC and CNF in the stabilization of Pickering emulsion [52]. They reported that the different sizes and flexibility of NC affected its rheological percolation and essentially its roles during emulsification. Both CNC and CNF were able to stabilize the emulsion well and form emulsion droplet clusters. However, their stabilization mechanism differed as shown in Figure 8A. In the case of CNC-stabilized emulsion, most short and rigid CNCs positioned themselves between the interfaces of two phases because of their small hydrodynamic size (0.1 µm) when compared to the size of the emulsion droplets (1–2 µm) formed. This positioning of CNCs promotes the repulsive effect and physical barrier between the emulsion droplets. At the same time, a small amount of CNCs act as a ‘bridge’, connecting the CNC-covered emulsion droplets and resulting in a percolation network. However, in the case of CNF-stabilized emulsion, steric hindrance between the CNF percolated network is superior and percolation networks are easily formed because of their low percolation threshold (0.12–0.13 wt.%) combined with the break-up and dispersion of oil droplets.

Meirelles et al. studied the stabilization mechanism of oil-in-water emulsion using CNC as stabilizing particles by measuring the interfacial tension of the emulsion [117]. By studying the interfacial tension between oil droplets and water, they found that CNC did not play a part in the reduction of the interfacial tension between water and oil. This indicates that the mechanism of CNC-stabilized emulsion is different from conventional emulsifiers, where the rod-shaped CNC flexibly surrounded the droplet surface and develop an effective interfacial coating. The adsorbed CNC then induces steric hindrance and electrostatic repulsion as a mechanism for droplet stabilization.

Bai et al. postulated that the good long-term stability of CNC-stabilized Pickering emulsions is associated with three stabilizing mechanisms [118]. First, the high surface potential of CNC (zeta potential of −55 mV) resulted in strong electrostatic repulsion between the oil droplets. Secondly, the formation of a thick CNC layer surrounding the droplet caused strong steric repulsion. Lastly, CNCs were irreversibly adsorbed onto the droplet surface. However, a slightly different stabilizing mechanism was suggested when the pH of the Pickering emulsion was adjusted to 2 as shown in Figure 8B. A limited amount of droplet flocculation was observed in this emulsion due to partial protonation of the sulfate ester groups, resulting in a weaker electrostatic repulsion than Van der Waals attraction. Coalescence was not observed as steric repulsion was still present in close range of the droplets.

Bai et al. explored the stabilization mechanism of Pickering emulsion by complexes of CNC and ethyl lauroyl arginate (LAE) [119]. At low LAE loadings, it was found that not all CNCs were absorbed onto the droplet interface due to the high negatively charged complex and droplets were stabilized by the partially neutralized CNCs. At intermediate LAE loadings, large oil domains were observed as a bilayer of admicelles formed on the surface of CNC, which resulted in a low surface charge and decreased effective coverage. Here, they postulated that with an increase in the LAE concentrations, the aggregated complexes might experience a transformation. However, at high LAE loadings, a bimodal system was shown, where CNC complexes adsorbed onto large droplets and several small droplets while the surfactant alone might also stabilize the remaining small droplets. All in all, they proposed that the stabilizing mechanism of Pickering emulsion was highly dependent on the type of complex formed, the absorbed surfactant on CNC, the presence of free surfactant, and the nature of oil.

Kalashnikova et al. studied various aspect ratios of CNC and the effects on its ability to form Pickering emulsion [120]. They found that all CNCs were able to be irreversibly adsorbed onto the emulsion droplets and formed an ultra-stable emulsion, which illustrated that they might have different stabilizing mechanisms. The different lengths of CNC affected their interfacial coverage, where a shorter CNC induced a more efficient surface coverage while the longer CNCs entangled with one another and bridged interconnecting droplets. At a low concentration domain, individual droplets can be obtained if neighboring droplets are absent at a distance relative to the length of CNC, where interconnections cannot be made. However, at a high concentration domain, CNCs would accumulate on a fixed interfacial area and form bridging networks for droplets, which is highly dependent on the length of CNC.

For CNF to stabilize fluid-–fluid interfaces, CNF particles from the aqueous bulk are required to be adsorbed onto the oil–water (O/W) interface. The particles must first diffuse into the vicinity of the interface, which is followed by positioning in the liquid–liquid boundary phase. The positioning of the particles in the liquid–liquid phase is highly dependent on the surface chemistry of the particles [74]. Mechanical treatment or a TEMPO-mediated oxidation process allowed the formation of CNF with various different physical and chemical characteristics. Thus, researchers studying the CNF-stabilized Pickering emulsion have suggested a slightly different mechanism for the stabilization process.

Silva et al. synthesized cationic-CNF with different degrees of cationization and aspect ratios from sugarcane bagasse [121]. They were then successfully utilized to prepare O/W Pickering emulsions, showing stability for up to 6 months with no macroscopic phase separation. Cationic CNFs were found to offer better colloidal stability in emulsion when compared to negatively charged TEMPO-oxidized nanocellulose. The interfacial stabilization by cationic-CNF can be attributed to dual stabilization mechanisms as shown in Figure 8C. The flexible fibrils first wrap themselves around the oil droplets and are adsorbed by them due to the strong electrostatic attraction with the oil surface. The remaining CNF that is not adsorbed onto the oil droplets then becomes entangled and forms strong networks in the vicinity of the continuous phase. This entanglement network increases the viscosity of the continuous phase and prevents the oil droplet from coalescence, which ultimately gives rise to a highly stable dispersion of oil droplets.

Lv et al. used CNF and chitin nanofibrils and their mixtures to form stable Pickering emulsions [122]. They suggested that the fibrils adsorbed onto the oil droplets and formed a thick particulate layer that shielded them from coalescence. After, the fibrils also formed a 3D network in the continuous phase, which prevented them from moving and forming a highly stable emulsion system. However, in this study, they postulated two possible physiochemical mechanisms by the fibril mixtures: multilayer or complex formation, as shown in Figure 8D. In the multilayer formation, anionic CNF first adsorbed on the oil droplet surfaces followed by the cationic chitin nanofibrils adsorbing on top of the cellulose nanofibrils, resulting in an overall net positive charge. In the complex formation, cationic chitin nanofibrils may have first adsorbed on the surfaces of the anionic CNF, forming fibril complexes, followed by the adsorption of the complex fibrils onto the oil droplets.

Nomena et al. investigated the mechanism of the stabilization of emulsions by cellulose microfibrils formed by de-agglomerated primary plant cell wall dispersions of citrus fiber [123]. They observed that the stabilization mechanism of the emulsion was attributed to both surface-active cellulose microfibrils and soluble polymers in the plant cells. From cryo-SEM images, it was observed that individual microfibrils surrounded the oil droplets while a bundle of fibrils was anchored to several droplets. These fibrils can spread along oil droplets or form flocs between the droplets. Moreover, the fibril bundles and the un-adsorbed fibrils formed a viscoelastic network surrounding the oil droplets, which minimized the movement of the droplets. On the other hand, soluble protein polymers can aid the adsorption of the fibrils onto the interface of the droplets and prevent coalescence by forming a shell. They also observed that the flocculation of oil droplets was attributed to the depletion attraction of soluble polymers and bridging by cellulose microfibrils.

However, Bai et al., on the other hand, employed CNF to induce depletion stabilization of oil-in-water Pickering emulsions formed by CNC through interfacial adsorption [124]. CNF in the aqueous phase formed flocs and induced a depletion effect, which was concentration dependent. At low concentrations (0.01 wt.%), CNF was able to induce depletion flocculation. However, at a concentration between 0.1 and 0.2 wt.%, they postulated that the emulsion was stabilized by depletion stabilization. The depletion stabilization was induced by a kinetic effect instead of a thermodynamic effect. The weak CNF network in this emulsion developed depletion stabilization by allowing the elastic properties of CNF to hinder the movement of the flocculated droplets. For CNF concentrations exceeding 0.3 wt.%, the stability of the emulsion was not caused by the depletion effects but by the formation of a strong CNF network within the continuous phase, restricting the movement of oil droplets.

## 4. Factors Affecting the Stabilization Profile of CNC- and CNF-Based Pickering Emulsion

As mentioned before, several crucial characteristics of a Pickering emulsifier affect the formation of a stable Pickering emulsion. In this section, the impact of several key factors on the stabilization performance of an emulsion and other non-major factors that may influence the emulsification-demulsification mechanism of water–oil interfaces are highlighted. Bio-based nanoparticles (CNC, CNF, and derivatives) are the main focus of this investigation to review their potential as a Pickering emulsifier.

### 4.1. Influence of the Hydrophilic–Hydrophobic Interfaces of Nanocellulose

The surface wettability of nanocellulose can be determined by the wetting contact angle using Young’s equation, with a hydrophilic particle having a lower contact angle while a hydrophobic particle has a higher contact angle. The degree of the hydrophobic or hydrophilic interface in nanocellulose will affect how they are absorbed between the interface of the two immiscible phases and determine the type of emulsion formed. Even though nanocellulose is amphiphilic in nature due to the difference in the crystalline allomorph, it is considered to generally be more hydrophilic. Thus, the wettability of nanocellulose can be altered by modification of the hydroxyl groups and this can be achieved by the adsorption or grafting of hydrophobic groups of chemical agents onto the surface of nanocellulose. Here, several papers on the impact of the wettability of CNC and CNF on its application in Pickering emulsion are summarized.

#### 4.1.1. Cellulose Nanocrystal

Du Le et al. described the modification of CNC with octenyl succinic anhydride (OSA) and studied its ability to stabilize oil-in-water emulsion in different pH and ionic strengths [125]. OSA was able to increase the hydrophobicity of CNC as evidenced in the increase in the water contact angle from 56° to 80.2°. The modified CNC was able to form stable emulsion droplets of 1.22 µm and showed no phase separation for 4 weeks under a refrigerated environment. These emulsions demonstrated aggregation under the conditions of pH < 4.0 and ionic strength ≥ 20 nM NaCl but resisted coalescence in the range of pH and ionic strength conditions studied. They suggested that the unique formation of elastic gel at low pH and in the presence of ions by the modified CNC can be applied as a carrier for bioactive compounds, where the emulsions showed resistance to coalescence and responsiveness to flocculation at bio-relevant pH and ionic strength. Another study by Chen et al. also prepared OSA-modified CNC and studied the fabrication of high-internal-phase emulsions. They found that even at low concentrations of modified CNC, a stable and gel-like emulsion with fine droplets was formed [126].

Gong et al. prepared hydrophobic CNC by oxidizing wood cellulose and functionalizing it with phenyltrimethylammonium chloride (PTAC) [127]. This modified CNC was homogenously stable in water and capable of stabilizing an oil-in-water emulsion better than a traditional surfactant, Tween-20. The modified CNC displayed a significantly better dispersed phase volume fraction than Tween-20 at the same concentration, resulting in a smaller quantity of modified CNC, which is sufficient to be used as a Pickering stabilizer compared to a traditional surfactant. Moreover, the emulsion stabilized by this modified CNC also exhibited better mechanical and thermal stability as seen by the presence of smaller emulsion droplets.

Tang et al. utilized the high surface charge density of sulphated-CNC and modified it by introducing polystyrene chains at the reducing end of the cellulose chains [128]. The modified CNC was then studied for its ability to form an emulsion with toluene and hexadecane and compared with the unmodified CNC. The modification process was successful as the hydrodynamic diameter was increased from 25.4 to 34.5 and 55.7 nm due to the aggregation induced by hydrophobic interactions. Modified CNC showed advantageous surface properties and better stability against coalescence for more than 4 months while the highly charged unmodified CNC-stabilized emulsion displayed coalescence and total phase separation immediately.

#### 4.1.2. Cellulose Nanofibril

Tang et al. altered the wettability of nanocellulose by introducing the hydrophobic domains cinnamoyl chloride (CC) or butyryl chloride (BC) onto CNFs [129]. They achieved this through an esterification reaction between CNFs and the hydrophobic group for 2 or 4 days. The hydrophobicity of the fibrils increased with the treatment time, and this can be observed by their wetting contact angle. The fibrils grafted with CC had a contact wetting angle of 51.43° and 68.36° for 2 and 4 days, respectively, while the un-modified CNF only had a contact angle of 46.10°. Both nanofibrils grafted with CC were then utilized to study the difference in the hydrophobicity and its effect on the characteristics of Pickering emulsion. In the hexadecane–water emulsion, both fibrils showed good stability and did not show coalescence of the droplets for over 5 months. However, in the toluene–water emulsion, the more hydrophobic nanofibrils (68.36°) formed smaller droplets than the other nanofibril (51.43°), with droplet sizes ranging from ~5 to 25 and ~32.5 to 35 µm, respectively.

Zhang et al. adjusted the hydrophobicity character of TEMPO-oxidized bacterial cellulose by initiating the adsorption of soy protein isolate nanoparticles on the surface of the bacterial nanofibrils through electrostatic attraction [130]. This modification was shown to slightly increase the hydrophobicity of the nanofibrils as shown by the slight increment in the wetting contact angle from 65° to 69° with an increasing weight ratio of the protein nanoparticles. This solid nanoparticle complex was able to form a highly stable oil–water Pickering emulsion with a droplet diameter of 10–40 μm. Additionally, the cellulose and protein complex created a physical layer between the two immiscible phases, which blocked the contact between the dispersed phase and the digestive enzymes and oxygen. Thus, this complex was found to be a suitable Pickering stabilizer for applications in functional food due to its refined anti-digestibility and oxidative stability.

Guo et al. investigated the role of residual lignin in nanocellulose for the application of Pickering emulsion [131]. Lignocellulose nanofibrils (LCNFs) with different lignin contents were prepared to stabilize oil-in-water emulsions. Two LCNF samples with a similar morphology and structure were synthesized by two pre-treatment steps (hydrothermal treatment and acid hydrolysis) followed by microfluidization of the fibers. The lignin content of the fibrils was controlled by the different temperatures of hydrolysis with p-toluenesulphonic acid. The LCNF-H fraction was labeled as having a higher lignin content (16.4%) and the LCNF-L fraction a lower lignin content (10.7%). The water contact angle of LCNF-H was found to be higher than LCNF-L, with 34° and 29° respectively, indicating that LCNF-H was more hydrophobic than LCNF-L. This could be due to the lignin being partially wetted by water and showing lower hydrophilicity. Consequently, LCNF-H adsorbed more strongly to the oil–water interface, with the repulsive forces between lignin preventing coalescence from occurring. This resulted in a 21% smaller droplet diameter when compared to LCNF-L at a concentration of 0.25 mg mL^−1^ and higher emulsion stability. All in all, the residual lignin content in nanocellulose can be altered to form different emulsions depending on the type of lignin or the concentration of the particles.

Sulbarán-Rangel et al. synthesized partially acetylated cellulose nanofibrils from *Agave tequilana* bagasse with different degrees of substitution of hydroxyl groups into ester groups [132]. The synthesized fibrils were then mixed with toluene at a concentration of 0.5% to form a stable emulsion, lasting for at least 43 days. The acetylation process drastically changed the physical characteristics of the fibrils depending on the degree of substitution. A high degree of substitution increased the wetting contact angle and caused the CNFs to be shorter and thinner, which is beneficial for the formation of a stable emulsion. From this study, it was found that all cellulose nanofibril fractions with different degrees of substitution were able to form stable emulsions due to the formation of a network structure by the fibrils. However, small and hydrophobic CNFs produced smaller droplet sizes (10–60 µm) while big and hydrophilic CNFs produced larger droplets (20–80 µm). A study by Xu et al. also found that acetylated CNFs were able to produce highly stable emulsions due to the increase in the hydrophobic character [133].Table 4 shows a summary of the scientific studies for easier comparison.

In short, the surface wettability of nanocellulose can be easily tuned via the physical or chemical method to increase the contact angle. This is carried out to improve the emulsifying ability and form smaller emulsion droplets. In theory, an emulsifying particle with a wetting contact angle close to 90° should perform the best. However, optimization of the modification process to achieve this has not yet been studied and reported. Thus, optimization of the modification process to synthesize nanocellulose with a wetting contact angle close to 90° should be the next step in research to fully make use of its potential.

### 4.2. Influence of Cationic, Anionic, and Neutral Phases of Raw or Modified Nanocellulose

Pickering emulsions stabilized by nanocellulose with a cationic or anionic surface have an influence on how the emulsion droplets behave. A net surface charge due to the cationic or anionic surface can produce electrostatic repulsion against the same charged surfaces or electrostatic attraction with the oppositely charged surfaces, affecting the duration of the emulsion stability. The surface charge density can be easily determined by its zeta potential value, with a high absolute value indicating a high surface charge density and vice versa. This value can be adjusted by incorporating either cationic, anionic, or neutral molecules onto the surface of the nanocellulose. Here, several studies on the influence of the surface charge density of CNC and CNF on its emulsifying ability are summarized.

#### 4.2.1. Cellulose Nanocrystal

The conventional practice in the preparation of CNC is by acid hydrolysis of cellulose fiber with sulfuric acid. This method allows the formation of negatively charged CNC due to the presence of sulfate groups on the surface of CNC. However, the degree of surface charge can be altered by controlling the acid hydrolysis parameters, which may expand their application. In a study by Saidane et al., they synthesized CNC with different sulfate surface charge densities (with a zeta potential of −30, −43, and −60 mV) by adjusting the concentration of sulfuric acid and stirring temperature during the hydrolysis process [2]. They found that the three different surface-charged CNCs caused coalescence and separation of both phases in a hexadecane-in-water emulsion in the absence of salt. However, when salt was added, the different CNCs were able to successfully stabilize the emulsion, with the droplet diameter remaining unchanged for more than a year. In the presence of salt, they also studied a styrene-in-water emulsion with the different CNCs and found that the size and particle size distribution of the larger beads/droplets were also similar when compared to the hexadecane-in-water emulsion.

In another study by Zhang et al., they altered the concentration of sulfuric acid during the hydrolysis process and synthesized sulphated CNC with different sulfur contents [134]. The sulphated CNCs were then studied for their ability to stabilize medium-chain triglyceride oil-in-water emulsion. The highest acid concentration used in this study (46.86 wt.%) was able to form CNC with a sulfur content of 1.499 mmol g^−1^ and a zeta potential of −47.98 mV while the lowest concentration used (22.72 wt.%) formed CNC with a sulfur content of 0.436 mmol g^−1^ and a zeta potential of −30.49 mV. They found that CNC with the highest sulfur content and surface charge led to the highest storage stability.

Pandey et al. observed the stabilization of a dodecane-in-water emulsion by CNCs with different degrees of surface charge [135]. The degree of surface charge was reduced by acid or basic desulfation of CNC, with the zeta potential being ≈−15 and ≈−25 from ≈−42.5 mV, respectively. They found that an aggregated lower surface charge of CNC (acid desulfation) had faster adsorption kinetics because of the lower electrostatic interaction between the particles and the oil–water interface, resulting in smaller emulsion droplets but a lower coverage ratio. On the other hand, CNC with a higher surface charge (basic desulfation) was less aggregated and formed a larger emulsion droplet and higher surface coverage ratio. However, in both emulsions, non-absorbed CNC formed a 3D network of particles in the continuous phase, which was attributed to the stability of the emulsion. Lower-surface-charged CNCs formed a stronger network because of the increased hydrogen bonding and van der Waals interactions.

Cherhal et al. also desulphated CNC by mild acid treatment with HCl but compared its ability to stabilize hexadecane–water emulsion with sulfated CNC [136]. They also found that the lower-surface-charged (uncharged) CNC showed aggregation and formed an 18 nm porous, heterogenous interfacial layer between the phases while the charged CNC showed an average thickness of 7 nm with respect to the size of its individual elementary crystals.

#### 4.2.2. Cellulose Nanofibril

The surface charge of CNF is usually altered through the TEMPO-mediated oxidation process of cellulose. In this process, the hydroxyl groups on the surface of nanocellulose are substituted into a negatively charged carboxyl group, which increases the surface charge density. Jimenez Saelices and Capron studied the importance of the surface charges of CNF in oil–water Pickering emulsion applications [137]. They studied a Pickering emulsion with TEMPO-oxidized CNF and high-pressure homogenized CNF with different surface charges. The negatively charged surface groups of TEMPO-oxidized CNF caused a higher degree of fibrillation because of the electrostatic repulsion, increasing its specific stabilized interface area. It was found that the TEMPO-oxidized CNF and CNC were able to produce individual non-creaming nano-sized droplets with an average diameter of 350 nm while the less fibrillated HP-homogenized CNF could only form micro-sized droplets of 5 µm, with the presence of several aggregated droplets.

In another study by Liu et al., they synthesized two TEMPO-mediated oxidized CNF from microcrystalline cellulose with different degrees of oxidation (52.8% and 92.7% were labeled as DO50 and DO90, respectively) to stabilize an emulsion with inhibited oil digestion [138]. The absolute zeta potential of nanocellulose DO90 was found to be higher than DO50, with a value of 59.4 and 50.8 mV, respectively, because of the increase in carboxyl groups on the surface of DO90. Through microscopic images, the emulsions by DO50 and DO90 CNFs formed droplets (around 15 µm) distributed within the solution. Through the Turbiscan stability analysis, they proved the excellent stable colloidal properties of the two emulsions when compared to un-oxidized cellulose. The negatively charged carboxyl groups increased the repulsion between the droplets, preventing the agglomeration and Ostwald ripening of the droplets. In addition, the fibrils also formed a 3D network surrounding the droplets, which acted as a barrier that prevented aggregation of the droplets.

Aaen et al. synthesized two types of CNF with different surface charge densities, one being low-charged enzymatically treated CNF (CNF-E) and the other being highly charged 2,2,6,6-tetramethylpiperidine-1-oxyl (TEMPO)-oxidized CNF (CNF-T) [139]. Highly charged CNF-T had a charge density of 1.49 mmol g^−1^ while low-charged CNF-E had a charge density of 0.044 mmol g^−1^. The two samples were then used to prepare oil-in-water Pickering emulsions with 40 wt.% rapeseed oil in the presence of sodium chloride (NaCl) and acetic acid, which is appropriate within the food industry. Both CNFs were able to form stable emulsions, which did not show coalescence and creaming after one month of storage. They also tested the stability of the emulsion after centrifugation and found that CNF-E-stabilized emulsion showed good stability with only a slight increase in the droplet size was observed in the presence of NaCl or acetic acid. Unlike the results reported by Liu et al., they found that the addition of acetic acid or NaCl caused the CNF-stabilized emulsion to become highly unstable [138]. This study indicated that the addition of NaCl led to the screening of negative charges on the CNF-T surface, causing irreversible aggregation of the fibrils under shear and lower viscosity of the continuous phase. In the case of addition of acetic acid, protonation of the carboxyl groups occurred and lowered the charge density of the fibrils. This observation showed the importance of electrostatic repulsion between the fibrils and the high viscosity for a highly stable emulsion.

Silva et al. synthesized two cationic CNFs with different degrees of substitution by glycidyl trimethylammonium chloride (GTMAC) [121]. The cationic CNFs were then compared with their anionic analogues to Pickering emulsifiers. The zeta potential value was increased from +24 to +37 mV when a higher molar ratio of GTMAC was used during the substitution process, showing a higher degree of substitution of GTMAC in the nanocellulose. Both cationic CNFs were utilized as Pickering emulsifiers at a concentration of 0.5 and 1 wt.% in a 30:70 oil:water ratio mixture. They exhibited outstanding stability against creaming and oiling off for a duration of 6 months whereas the anionic analogue showed phase separation in the emulsion after 24 h of storage under the same conditions. This could be due to the electrostatic attraction between the cationic CNF and the negatively charged immiscible phase while the anionic analogue experienced electrostatic repulsion with the oil phase, leading to early phase separation. Table 5 compiles a summary of the scientific studies for easier comparison.

Briefly, the surface charge density of nanocellulose affects how the emulsion droplets behave and the emulsion stability. As mentioned earlier, sulfuric acid hydrolysis is commonly used to synthesize negatively charged CNC whereas TEMPO-mediated oxidation is usually performed to impart a negative surface charge for CNF. However, these chemicals are considered to be dangerous, with sulfuric acid being extremely corrosive and TEMPO being toxic. Therefore, more research should be carried out on the use of safer chemicals or methods to not only synthesize nanocellulose but also impart surface charge as well.

### 4.3. Morphology

CNC has a rod-like structure with a short length of 100 to several micrometers whereas CNF has a flexible and entangled structure with a length of more than 10 µm. Their difference in morphology determines how they are positioned between the phases and affects the coverage ratio of droplets. Their contrast in sizes also has an effect on the stabilizing mechanism of Pickering emulsion as mentioned in Section 3. Here, several studies on the different morphology of CNC and CNF on their ability to form stable emulsions are summarized.

#### 4.3.1. Cellulose Nanocrystal

Ni et al. utilized high-pressure homogenization as a post-treatment after the synthesis of CNC to reduce the length and found that the length of CNC decreases with the increase in the homogenization pressure [83]. They prepared Pickering emulsions with a 50% oil phase and 0.15% CNC to determine the effect of the length of nanocellulose on the stability of the Pickering emulsion and observed that the reduced particle sizes decreased the emulsion droplet size while increasing the surface coverage ratio. In addition to this, they also found out that a small amount of CNC treated with 50 MPa pressure was able to stabilize the emulsion, with the oil phase ranging from 10 to 70 (*v*/*v*) %, and the environmental factors (temperature, ionic strength, and pH) did not affect the stability of the emulsion.

Kalashnikova et al. investigated the effect of the aspect ratio of CNC on oil-in-water Pickering emulsion [120]. Three types of CNCs were obtained from the hydrolysis of cotton, bacterial cellulose, and *Cladophora*, with aspect ratios of 13, 47, and 160, respectively. All three CNCs showed a good ability to form ultrastable emulsions with similar droplet sizes under diluted conditions. However, they found that the aspect ratio of CNC plays a crucial role in the coverage ratio of droplets. A shorter CNC was able to display a dense organization around the droplet (coverage > 80%) while the longer CNC formed an interconnected network surrounding the droplets (coverage ≈ 40%).

Wang et al. prepared differently sized CNC by varying the duration of the acid hydrolysis process [140]. In this study, CNC with the shortest length (178.2 nm) was synthesized with a hydrolysis duration of 3.5 h while the longest CNC (261.8nm) underwent a hydrolysis process of 1.5 h. All prepared CNCs were then used to study their ability to form stabilized palm oil/water emulsion. The results indicated that the CNC with a smaller particle size had a higher emulsification efficiency as seen by the increase in the emulsion layer. The addition of casein was observed to improve the emulsion, with an increase in its concentration. However, the addition of salt affected the emulsion differently, where a low NaCl concentration (0.1%) increased the emulsion volume; however, a higher NaCl concentration (≥0.2%) decreased the emulsion volume but decreased the emulsion droplets.

#### 4.3.2. Cellulose Nanofibril

CNF is generally considered to be long and non-uniform, and its morphology can be easily altered by the treatment process of cellulose and the severity of the process. Li et al. prepared bacterial cellulose nanofibrils (BCNFs) with varying physical sizes through high-pressure homogenization [84]. The nanofibrils were then used as stabilizers in oil-water Pickering emulsion. The average width of the fibrils decreased from 127 to 97 nm when the homogenized time increased from 10 to 80 times at a pressure of 750 Bar, indicating the significant influence of the homogenization process on the size of the fibrils. All the BCNF fractions showed a gel-like behavior in the emulsion, proving that a strong network surrounded the droplets. They observed that a more stable emulsion could be obtained by smaller BCNFs or at a higher concentration of BCNFs. Moreover, they also found that environmental factors such as the temperature, pH, and ionic strength had negligible effects on the stability of the emulsion. In short, the physical size of the fibrils affected the stabilization of the emulsion by forming intermolecular crosslinks and increasing the steric hindrance within the 3D network.

Tang et al. synthesized hydrophobically modified cellulose nanofibrils with different sizes by performing acid hydrolysis with different durations on cinnamoyl chloride (CC) and butyryl chloride (BC) grafted fibrils [129]. Among all the modified nanofibrils, the two shorter CNFs (29.39 and 58.77 nm average diameter) were able to stabilize toluene– or hexadecane–water emulsions, with a decrease in the droplet size and an increase in the particle concentration. On the other hand, the other two synthesized nanocelluloses showed longer diameters (295.3 and 575.8 nm), significantly affecting the stabilizing mechanism and the distribution of emulsion droplets. The longer fibrils showed a slower creaming rate in toluene–water emulsion due to the increment in the viscosity in the emulsion in the form of a 3D inter-particle network. However, they bridged multiple emulsion droplets with varying sizes in the hexadecane–water emulsion, resulting in a bi-modal distribution of the size of the droplets.

Wu et al. developed an acid-free process to prepare CNFs from kelp for oil/water Pickering emulsion applications [141]. CNFs were prepared via enzymatic treatment with cellulase and chemical treatment with TEMPO-oxidized CNF, respectively. On the one hand, cellulase-treated CNF showed long, entangled, and non-uniform fibers, with lengths of more than 3 µm. On the other hand, TEMPO-oxidized CNF had individual, uniform fibers, with lengths reduced to 0.6–1 µm and a width of 10–20 nm. In the Pickering emulsion study, TEMPO-CNFs showed a better performance than cellulase-treated CNFs as the oil droplet size was smaller ranged and from 20 μm to <10 µm. In addition, delamination of the emulsion was observed in the TEMPO-CNF emulsion after 3 days of storage, whereas the cellulase-treated CNF emulsion showed significant delamination under the same storage conditions. Table 6 compiles a summary of the scientific studies for easier comparison.

In brief, both CNC and CNF were found to be able to form stable emulsions with small emulsion droplet sizes, but the difference in their morphology only affected the coverage ratio of emulsion droplets. The difference in their sizes also affected the stabilization mechanism of the Pickering emulsion, as discussed earlier in Section 3. Thus, different sized nanocellulose such as amorphous nanocellulose, cellulose nanoyarn, cellulose platelets, and hairy cellulose nanocrystalloids should be studied for their effectiveness as a Pickering emulsifier.

### 4.4. Influence of Other Non-Major Factors

Although the characteristics of nanocellulose/derivatives (wettability, surface charge, and particle size structures) are the main criteria for the effective formation of Pickering emulsion, there are other aspects that indirectly change the stabilization mechanism of CNC- and CNF-stabilized Pickering emulsion stabilizer. These unexpected phenomena are due to the presence of different physicochemical properties of nanocellulose that are altered under processing conditions or external factors found within the emulsion mixture that affect the effectiveness of the nanocellulose as a Pickering emulsifier. Here, we summarize studies on the non-major factors affecting the stabilization profile of CNC and CNF Pickering emulsion.

#### 4.4.1. Cellulose Nanocrystal

Li et al. investigated the effect of crystalline allomorph of CNC as effective stabilizers for Pickering emulsion applications [142]. They prepared two different crystalline allomorph CNCs by sulfuric acid hydrolysis of unmodified and mercerized microcrystalline cellulose, namely CNCs-I and CNCs-II, respectively. CNCs-I had a needle-like particle size with a length of 200 nm and a width of 16.4 nm while CNCs-II had an ellipsoid shape with a length of 18.8 nm and a width of 10.9 nm. CNCs-I displayed a superior emulsifying ability, with a larger emulsion ratio and smaller emulsion droplets than CNCs-II. This shows that the difference in the crystalline allomorph affected the morphology and wettability, which ultimately affected the emulsifying efficiency.

Liu et al. reported a protein-covered CNC as an effective stabilizer of high internal phase emulsion [143]. This was possible due to the introduction of protein to the CNC surface, which significantly affected the amphiphilic properties. The formation of a gel-like emulsion was attributed to the sticky protein particles being closely packed together and the creation of a 3D network in the emulsions. They also found that by altering the surface coverage of CNCs or the concentration of protein-covered CNC particles, the stiffness and microstructure of the emulsion could be modulated.

Complexes of CNC and lauric alginate surfactant formed by electrostatic attraction were studied for their application in food emulsion [144]. The complexes were able to form stable oil-in-water emulsions with small emulsion droplets. The formation of the highly stable emulsion was due to the formation of the coating by the complex providing strong electrostatic repulsion while the formation of small emulsion droplets was caused by the rapid adsorption of the surfactant, respectively. The emulsion formed by 0.02% CNC and 0.1% surfactant was able to resist droplet coalescence and extend the lag phase to produce lipid hydroperoxide and hexanal for 20 and 14 days, respectively.

#### 4.4.2. Cellulose Nanofibril

Souza et al. studied the influence of the essential oil chemical structure on a CNF-stabilized Pickering emulsion [145]. In this study, three different essential oils were used, namely, cinnamon cassia, cardamom, and Ho wood. The average diameter of the droplets formed in the three emulsions ranged from 10 to 20 µm depending on the surface coverage of CNF. CNF-stabilized cardamom oil showed the highest surface coverage value at 24.6% and Ho wood oil was the lowest at 20.6%. They found that the coverage value reflected the stability of the emulsion and the diameter of the droplet formed, with the emulsion with cardamom and cinnamon oil being stable whereas the emulsion with Ho wood oil being unstable after 14 days. From the FT-Raman analysis, they postulated a correlation between the chemical structure of essential oil and the emulsion stability. Cinnamon and cardamom had stable chemical structures and resulted in steric stabilization. Ho wood, on the other hand, had a reactive chemical structure that was able to form hydrogen bonding with CNF. This resulted in the formation of a monolayer around the oil droplets and caused coalescence of the droplets.

Xie et al. utilized a cellulose nanofibrils (CNFs)/carboxymethyl chitosan (CCS) mixture as a Pickering stabilizer in beeswax–water emulsion as an edible coating film [146]. The emulsion by the mixture had a uniform droplet distribution of around 10µm. By increasing the contents of CNF in the mixture, the emulsion displayed smaller droplets, a narrower droplet size distribution, and better creaming stability.

Buffiere et al. compared two types of nanocelluloses obtained from near-critical water (250–300 °C) and high-shear homogenization of microcrystalline cellulose (MCC) as Pickering stabilizers [147]. Both processes were able to form nanocellulose but with different morphology and amphiphilic properties. The near-critical water treatment was able to depolymerize MCC efficiently and formed a low molecular weight that consisted of cellulose II, whereas the homogenization process disassembled MCC without affecting its molecular structure. The near-critical-water-treated nanocellulose displayed a better performance as a Pickering stabilizer as it was able to form stable emulsion at a concentration as low as 1.0 wt.% while the micro-fibrillated cellulose from high-shear homogenization required a concentration of 5.0 wt.%.

Lv et al. fabricated a stable Pickering emulsion using either cellulose and/or chitin nanofibrils (CNFs and ChNFs), respectively [122]. The emulsions formed were found to be highly stable against coalescence and creaming during storage. In the study, they were able to identify the optimal fibrils concentration as well as their ratio to form a stable emulsion. The lowest fibril concentration required for a stable emulsion with a small oil droplet size (1–10 µm) was determined to be 0.3%. At a concentration of 0.3%, the most studied ratio showed negligible change in the droplet size during storage except for the ratio of 2:1 of CNF:ChNF, where a decrease in the droplet size was observed after 7 days of storage. They postulated the breakdown of flocs through the rearrangement of the different fibrils, minimizing the bridging flocculation.

From the aforementioned studies, it is clear that there are many factors that can affect the Pickering emulsifying ability of nanocellulose. Other factors include the pH of the emulsion mixture, the addition of salt, the type of oil, and the oil:water ratio. It is important to note that all these factors may indirectly alter the morphology, surface charge, and wettability of nanocellulose, which will then affect their effectiveness in stabilizing Pickering emulsion. Therefore, all these factors must be considered when utilizing nanocellulose as a Pickering emulsifier with the intention of incorporating them into consumer products within the food, cosmetic, and pharmaceutical industries.

## 5. Toxicity and Safety Aspects of NC

According to the Food and Drug Administration (FDA), cellulose is generally considered to be a safe food substance while the EU regulations recognize it as a food additive [148]. Cellulose fibers have been added to food products to provide structure, texture, and positive sensory effects.

However, nanotechnology is in the development stage. Clinical trials, toxicity, and biodegradable-related studies on nanoparticles applied to the human body remain scarce. Thus, nanocellulose is considered to be entirely different compared to cellulose because nanomaterials may behave differently or have unexpected properties compared to their bulk material. As mentioned before, the variety of advantageous properties in nanocellulose has caused an upsurge in studies of its use in research and industry over the past years. This will inevitably lead to an increased release of nanocellulose into the environment [149]. In addition, various types of nanocellulose modifications (physical adsorption or chemical grafting) to alter its wettability or surface charge must also take into consideration that these functional groups may interact with the cell membrane, resulting in downstream biological responses [150]. Its danger or potential risk to the environment and human health must be sufficiently studied before its implementation in commercial goods.

From Figure 9, the toxicity profile of nanomaterials is identified and analyzed for any possible risk scenarios throughout its life cycle. NANO LCRA is based on the principle of traditional risk assessment, which characterizes the risks of nanomaterials from the raw material stage to the disposal and end-life of the product [151]. From the flowchart, we can see the life cycle of any manufactured product containing nanomaterials, where several stages of nanocellulose exposure could occur. They can be identified as Stage 1, the production of raw materials; Stage 2, the manufacturing of sold goods; Stage 3, transportation of the sold goods; Stage 4, consumption by the consumer; and, lastly, Stage 5, disposal [151]. Of course, it is important to note that this life cycle can be used as a standard for the use of nanocellulose in many different fields, but other factors might need to be considered in other applications as well. The major exposure route will occur within one or two of the stages in the life cycle but will differ depending on the application or fields.

Studies on the toxicity of nanocellulose have yielded mixed results, with some studies indicating its benign characteristic and some showing otherwise. The difference in the results obtained could be caused by the toxicity tests (variation in cell systems, cell exposure doses, and exposure route), physical characteristics of nanocellulose (morphology, surface chemistry, and colloidal stability), extraction methods, and the origin of cellulose [59,152].

Torlopov et al. studied the acute toxicity of the different morphology of partially acetylated CNC and sulfated CNC [153]. Oral administration of CNC hydrosols to mice showed low acute toxicity, with an LD_50_ > 2000 ppm. In addition to this, it showed good hemocompatibility and did not cause platelet aggregation or destroy red cell membrane. Intravenous administration also did not affect the plasma clotting time in mice.

Soo Min et al. performed cytotoxicity tests, eye irritation tests, and skin irritation tests on cellulose nanofibers (CNFs) to prove their applicability in the cosmetics industry [154]. From the cytotoxicity test, CNF was found to significantly inhibit human skin keratinocyte cell and human dermal fibroblast cell growth when compared to the negative control. On the other hand, eye irritation and skin irritation tests were carried out and it did not cause irritation to the eye or skin at a concentration of 5000 µg/mL. All studies performed used a short-term 24-h exposure duration so long-term studies need to be performed to confirm its safe use in the cosmetics industry.

Harper et al. studied the relative influence of the aspect ratio and surface chemical charge on the behavior and toxicity of nanocellulose (CNC and CNF) using embryonic zebrafish as a vertebrate model of toxicity [155]. Their study found that surface chemistry had an insignificant influence on the toxicity, but a higher aspect ratio of CNF was more toxic than CNC in some cases. They postulated that the difference in the mechanical homogenization process and the origin of the material may have caused a difference in the aspect ratio, which eventually impacted the toxicity.

DeLoid et al. determined the toxicity of ingested nanocellulose (CNC and CNF) in physiologically relevant in vitro and in vivo systems [156]. From the findings, it can be concluded that the ingested NC has little acute toxicity and is mostly non-hazardous when consumed in small quantities. However, the long-term effects, potential detrimental effects on the gut microbiome, and the absorbance of essential micronutrients need to be studied to ensure their safe consumption.

Ogonowski et al. studied the toxicity of CNF for aquatic life using standard ecotoxicological tests and feeding experiments in Cladocera *Daphnia magna* [157]. They monitored the food uptake, growth, reproduction, and survival of *Daphnia magna* according to exposure to a CNF concentration ranging from 0.206 to 2060 mg L^−1^. Over the studied range of concentrations, no mortality was observed after exposure for 2 days. Moreover, a stimulatory effect on growth was observed after exposure to low food and moderate CNF levels but low food levels and the highest CNF concentration levels showed a negative effect on growth and reproduction. In short, they concluded that CNF showed a low toxicity profile to the environment and filter-feeding organisms.

From the several aforementioned studies, most of the results showed a low toxicity profile with short-term exposure. However, a multitude of factors need to be considered since the morphology and surface properties of nanosized materials affect their toxicity. The various types of differently synthesized nanocellulose will possess different toxicity profiles [158]. Moreover, the various surface modifications of NC also performed with different types of functionalization may impact their toxicity as well.

Regardless of the low toxicity profile of nanocellulose from the short-term exposure reported, the use of nanomaterials in the food, cosmetics, and pharmaceutical industries remains a problem as there are no regulations for their production, application, and means of disposal. Many countries, however, have respective guideline legislation and regulatory bodies to manage or control the potential risk carried by nanomaterials. In Europe, there is a regulation stating that nanomaterials, among other things in cosmetic products, must be included in the table of contents, but in the case of food products, they are evaluated individually [159]. An article by Rauscher et al. compiled and reported the regulatory aspects of nanomaterials in the EU [160]. In the US, regulatory agencies such as the Food and Drug Administration (FDA), United States Environmental Protection Agency (USEPA), and the Institute for Food and Agricultural Standards (IFAS) have introduced protocols pertaining to the possible risks of nanomaterials and products [161]. Still, more work must be carried out in accordance with the regulatory requirements for its safe use in consumer products and to improve our understanding of nanocellulose and nanomaterials.

## 6. Conclusions and Future Aspects

With the rapid growth of scientific and technological interest in nanocellulose from academic and industry players for non-toxic based applications, more studies have been focusing on nanocellulose’s usage as a Pickering emulsifier that meets the aspects of sustainable, eco-friendliness, safety, edibility, and biocompatibility for food, beauty, personal care, and hygiene-based products. In addition, the stabilization mechanisms of CNC and CNF in Pickering emulsion have been well established by researchers. In general, the stabilization mechanism of CNC follows the irreversible adsorption of the particles to form an interfacial film, which causes a steric hindrance between the droplets. For the CNF-based stabilization mechanism, the preferable dispersion pattern is via the formation of a 3D network that envelopes the emulsion droplets, which inhibits their movement. However, the stabilizing mechanism profile for other types of nanocellulose such as amorphous nanocellulose and hairy cellulose nanocrystalloids should be studied to determine their potential emulsifying efficiency.

The unique tunable characteristic of nanocellulose can be easily surface modified to alter their surface wettability, surface charge, and morphology. This makes nanocellulose materials versatile as an emulsifier for a wide range of formulations. Most of the studies reported in Section 4 by other researchers have shown their potential use in non-toxic Pickering emulsion (food, cosmetic, and biomedical) and improved emulsifying capabilities either by surface modification of nanocellulose or other methods. However, in these studies, their toxicity is not reported, which might be a concern for their use in consumer products.

Overall, modified or unmodified CNC or CNF were found to exhibit an excellent emulsifying ability as potential stabilizers in the food, cosmetic, and biomedical fields. Although nanocellulose is derived from renewable and natural-based plant/agricultural resources, the nanoscale dimension of nanocellulose may imply different biological effects, especially for human usage/consumption. In addition, nowadays, consumers demand safety information about the product used such as the daily dosage, nutrition profile, safe ingredients, calories per meal, and product expiry date. Thus, the full life cycle of risk assessment for nanocellulose-based products is a crucial process prior to marketing of the product. This pre-commercial screening involves the characterization of potential risks and toxicology testing (in vitro or in vivo) from raw materials through to the end-of-life or disposal/reuse, which helps to identify occupational, consumer, and environmental risks. Studies have summarized that nanocellulose does not seem to have any toxic effects either in vitro or in vivo. The physical characteristics (size, shape, surface charge), realistic dose, and exposure scenarios of nanocellulose are the criteria that limit the toxic potential. Lastly, more studies on the toxic effects of nanocellulose and its derivatives are required to provide a firm conclusion regarding the human safety aspect before their implementation in consumer products.

## Figures and Tables

**Figure 1 molecules-27-07170-f001:**
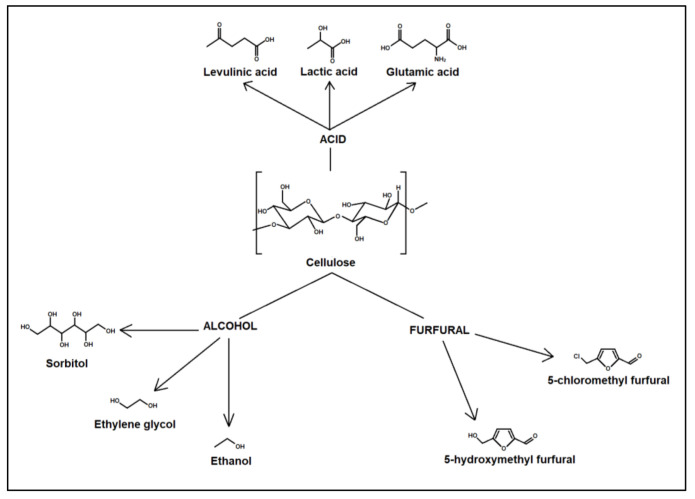
Platform chemicals that can be produced from cellulose.

**Figure 2 molecules-27-07170-f002:**
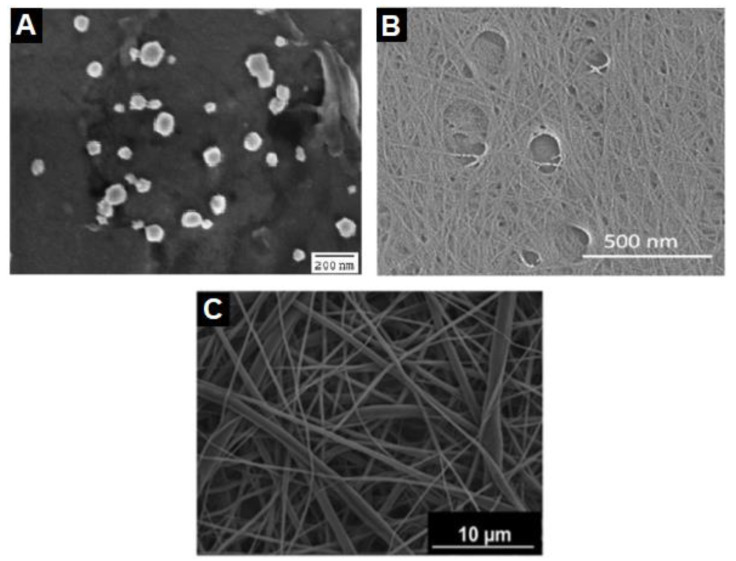
(**A**) SEM image of amorphous nanocellulose. Reproduced with permission from [54], [Carbohydrate Polymers]; published by [Elsevier, 2007]. (**B**) SEM image of cellulose nanoplatelets. Reproduced with permission from [55], [Carbohydrate Polymers]; published by [Elsevier, 2018]. (**C**) Top-down SEM image of electrospun cellulose nanofiber. Reproduced with permission from [57], [Carbohydrate Polymers]; published by [Elsevier, 2019].

**Figure 3 molecules-27-07170-f003:**
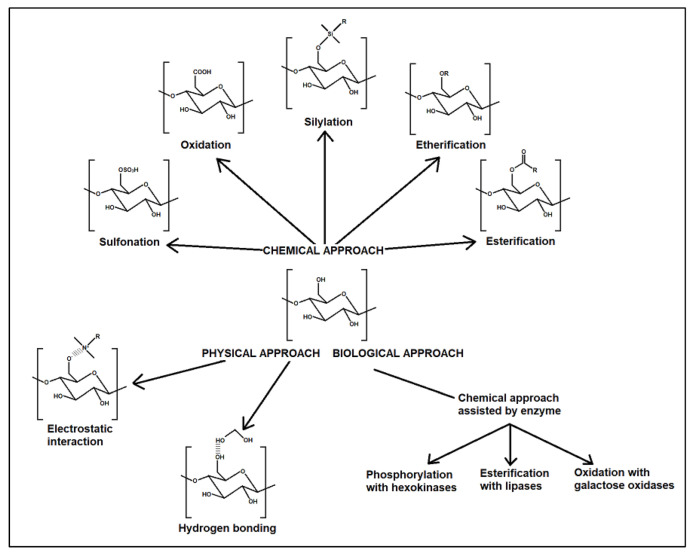
Several surface modification reactions of nanocellulose via the physical, chemical, and biological approaches.

**Figure 4 molecules-27-07170-f004:**
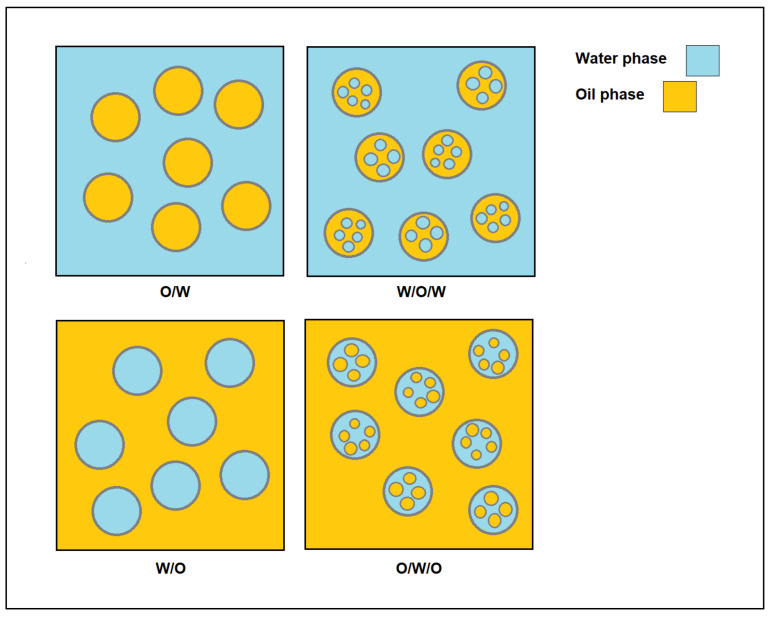
Schematic diagram of the different types of emulsions formed between water and oil.

**Figure 5 molecules-27-07170-f005:**
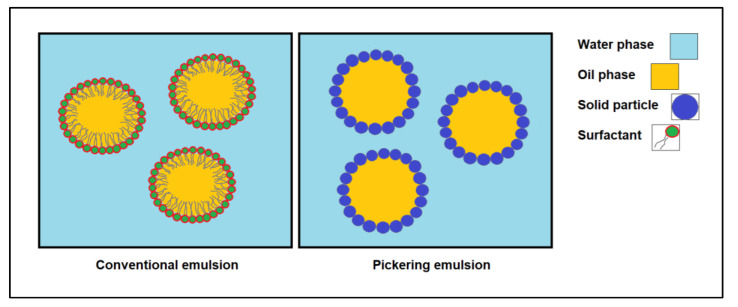
Schematic diagram of the structure of conventional emulsion and Pickering emulsion.

**Figure 6 molecules-27-07170-f006:**
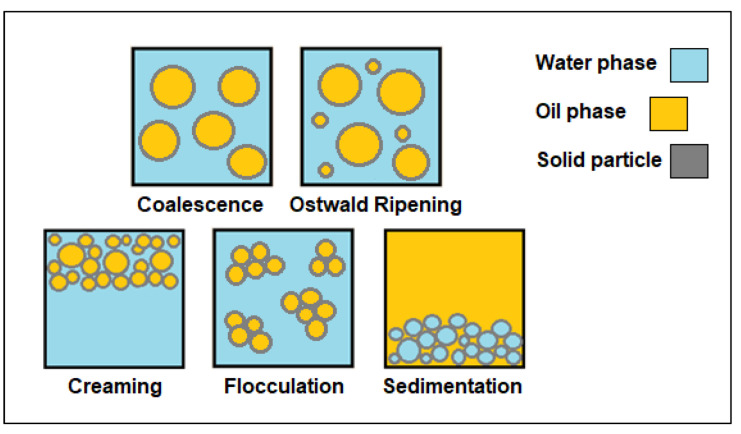
Schematic diagram of the different physical instability of the emulsion.

**Figure 7 molecules-27-07170-f007:**
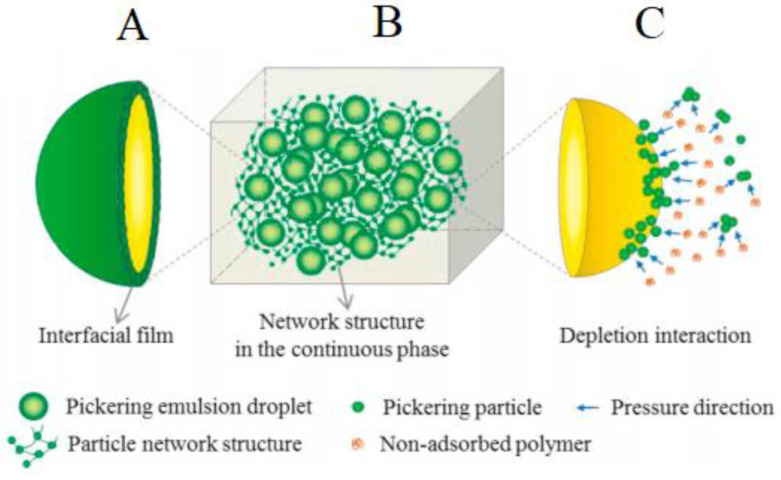
Schematic representation of the three possible stabilization mechanisms of Pickering emulsions: (**A**) formation of the interfacial film; (**B**) formation of a 3D network structure; (**C**) stabilization by depletion interaction. Reproduced with permission from [71]. [Food Hydrocolloids]; published by [Elsevier, 2021].

**Figure 8 molecules-27-07170-f008:**
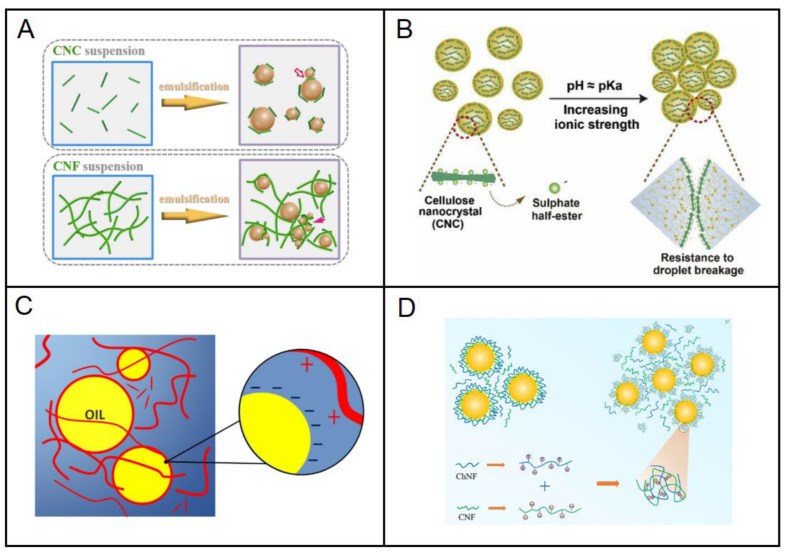
(**A**) Schematic representation of CNC and CNF suspension along with the olive-oil-in-water emulsion stabilized by CNC and CNF. Reproduced with permission from [52], [Carbohydrate Polymers]; published by [Elsevier, 2021]. (**B**) Schematic illustration of CNC-stabilized corn-oil-in-water emulsion under different pH environments. Reproduced with permission from [118], [Food Hydrocolloids]; published by [Elsevier, 2019]. (**C**) Schematic diagram of the proposed dual stabilization mechanism of cationic CNF-stabilizing almond-oil-in-water emulsion. Reproduced with permission from [121], [Journal of Colloid and Interface Science]; published by [Elsevier, 2020]. (**D**) Schematic representation of 2 possible stabilization mechanisms of CNF and chitin nanofibrils mixture stabilizing corn-oil-in-water emulsion. Reproduced with permission from [122], [Food Hydrocolloids]; published by [Elsevier, 2020].

**Figure 9 molecules-27-07170-f009:**
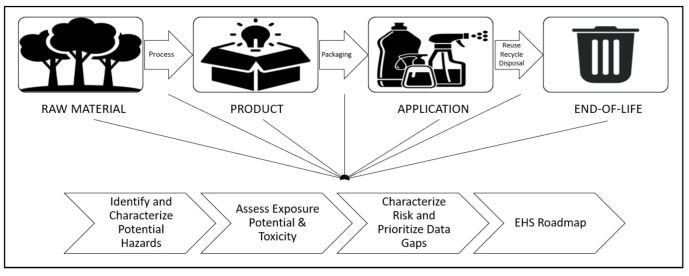
NANO LCRA flowchart of the complexity of risk assessment of a nanomaterial throughout its life cycle. Adapted from [151].

**Table 1 molecules-27-07170-t001:** Chemical composition of a variety of lignocellulosic biomass (% dry weight).

Source of Biomass	Cellulose (%)	Hemicellulose (%)	Lignin (%)	Others, (Extractives, Protein) (%)	Reference
**Agro-industrial waste**					
Hazelnut shell	30	23	38	3.74	[10]
Extracted olive pomace	19	22	40	35.29	[10]
Corncob	43.09	35.42	12.85	3.85	[11]
Walnut shell	20.47	20.16	45.93	6.22	[11]
Sugarcane bagasse	43.65	29.29	20.63	-	[12]
Palm oil frond	37.32	31.89	26.05	2.25	[13]
Rice straw	36.4	20.4	14.3	-	[14]
Coffee husk	35.4	18.2	23.2	17.8	[15]
Chili post-harvest residue	39.95	17.85	25.32	-	[16]
Wheat straw	40.10	32.93	18.39	10.60	[17]
Date palm rachis	47.31	25.72	15.67	5.80	[18]
**Grasses**					
Rye	42.83	27.86	6.51	-	[19]
Silage	39.27	25.96	9.02	-	[19]
Elephant grass	35.97	22.43	20.77	14.39	[20]
Napier grass	47.1	31.2	21.6	-	[21]
**Hardwood**					
Rubber wood	39.56	28.42	27.58	1.98	[13]
Poplar	46.74	31.73	23.92	3.89	[17]
Zeen oak	41.39	29.66	19.37	6.30	[18]
*Encalyptus globulus*	44.9	28.9	26.2	-	[22]
Hornbeam	34.2	17.0	26.3	-	[23]
**Softwood**					
Aleppo pine	37.32	30.48	25.04	6.97	[18]
Japanese cedar	52.7	13.8	33.5	-	[22]
Spruce wood	39.01–42.51	34.98–35.30	23.69–26.06	1.04–1.68	[24]
*Pinus radiata*	38.4–41.7	28.9–29.5	26.5–29.3	1.84–2.11	[25]
*Pinus pseudostrobus*	42.98	23.55	28.94	5.11	[26]

**Table 2 molecules-27-07170-t002:** Comparison of the different types of nanocellulose.

Types of Nanocellulose	Morphology	Properties	Synthesis Method
Cellulose nanocrystal (CNC), cellulose nanocrystalline, celulose nanowhiskers	- short rod-like or whisker shape - 2–20 nm in diameter - 100 nm to several micrometers in length	Strengths: - high surface area - high crystallinity - high strength and hardness Limitations: - low thermal property - low yield	- Acid hydrolysis - Enzymatic hydrolysis - Ionic liquid treatment - Deep eutectic solvent-based treatment - Combination of either 2 methods
Cellulose nanofibril (CNF), cellulose microfibril, microfibrillated cellulose, nanofibrillated cellulose	- long, flexible, and entangled - 20–100 nm in diameter - 10 µm in length	Strengths: - high aspect ratio - unique rheological properties Limitations: - poor mechanical properties - low crystallinity	- Mechanical treatment (homogenization, grinding, cryocrushing, extrusion)
Bacterial cellulose (BC), microbial nanocellulose	- pure, ultrafine, and ribbon-shaped - 30–50 nm in diameter - micrometers in length	Strengths: - high purity - great flexibility - high water absorption capacity - higher amount of hydroxyl groups Limitations: - High cost and low productivity of synthesis	- Cultured from bacterium (static, agitated, and bioreactor cultures)
Amorphous nanocellulose *	- spherical to elliptical shape - 50–200 nm in diameter/length	Strengths: - high accessibility - improved sorption - high degree of functionality Limitations: - poor thermal stability and mechanical properties	- acid hydrolysis followed by ultrasound disintegration
Cellulose nanoyarn, electrospun cellulose nanofiber *	- mats of tangled long filaments - 100–1000 nm in diameter - 10 µm in length	Strengths: - highly porous Limitations: - low mechanical characteristics and thermal stability	- electrospinning of cellulose solution
Cellulose platelets *	- entangled cellulose nanofibrils - 2–3 nm in diameter - 80 nm in thickness	Strengths: - large aspect ratio	- acidified oxidation - acid hydrolysis
Hairy cellulose nanocrystalloids *	- rod-shaped dimension with protruding chains on both ends - 5–10 nm in diameter - 100–200 nm in length	Strengths: - unique surface properties - high mechanical properties and colloidal stability - high degree of functionality	- periodate oxidation - chlorite oxidation

* Other types of nanocellulose that are not listed in the international standard, TAPPI.

**Table 3 molecules-27-07170-t003:** Summary of some recent research articles on the use of nanocellulose as a Pickering emulsifier in food, cosmetic, and biomedical applications.

	Application of NC	Nanocellulose Content in Emulsion	Oil Phase Type and Ratio	Droplet Size	Research Findings	Ref.
Food	CNC and CNF from lemon seeds co-stabilized sunflower oil Pickering emulsion	1 wt.%	Sunflower oil O/W (1:1)	60–160 µm in droplet size	- concentration and ratio of CNF in CNC/CNF mixture can alter the properties of the emulsion - the increase in the CNF concentration at a fixed CNC concentration (0.5 wt.%) improved the emulsion stability - environmental factors (pH and ionic strength) improved the storage stability	[80]
	Pickering emulsion of corn oil stabilized by different lengths of CNC from ginkgo seed shells	0.025–0.25% (*w*/*v*)	Corn oil O/W (1:9; 3:7; 5;5, 7:3)	1–50 µm in droplet size	- the longer CNC had a lower coverage ratio when compared to the shorter CNC - a small amount of CNC homogenized at 50 MPa can stabilize the high oil phase in the long term - Environmental factors showed negligible change in the stability of emulsions.	[83]
	Oil-in-water Pickering emulsion stabilized by different lengths of BC	0.1–0.5 wt.%	Dodecane O/W (1:9; 2:8; 3:7; 4:6; 5:5)	10–40 µm in droplet size	- the size of BC showed no obvious effect on the emulsifying capacity - the ratio of oil and BC significantly affected the surface coverage ratio of the emulsion droplets - environmental factors showed little influence on the emulsion stability	[84]
	Pickering emulsion of olive oil by different flexibility of NC	0.07–4.0 wt,%	Olive oil O/W (2:8 and 3:7)	-	- flexible BC showed almost no emulsifying capacity while CNC and CNF exhibited good emulsifying capacity - in the CNF-stabilized emulsion, steric hindrance is more dominant - in the CNC-stabilized emulsion, the repulsive effect is more important	[52]
	CNF from *Miscanthus floridulus* straw as Pickering emulsifier	0.05–0.20 wt.%	Dodecane O/W (1:9)	~10 µm in droplet size	- the emulsion stabilized by CNF showed a small droplet size and high stability - environmental factors have an insignificant influence on the emulsion stability - concentration of CNF affected the oil–water interface ≈ on the associated intermolecular cross-linking or steric hindrance	[95]
Food packaging	Poly (lactic acid)/CNC composite via Pickering emulsion	0.0017% (*w*/*v*)	Dichloromethane O/W (1:2)	-	- improved the rheological and thermal properties - increased storage and flexural modulus - homogenous dispersion of CNC in the composite	[86]
	Whey protein isolate files with bergamot oil emulsified by CNC	1.6 mg/mL 0.0016% (*w*/*v*)	Bergamot oil O/W (2:8)	-	- enhanced mechanical resistance and water vapor permeability - more homogenous structure - sustained antioxidant and antimicrobial activity - decreased resistant to tension	[87]
	Starch film containing essential oils emulsified by CNF	0.15 wt.%	Cardamom, cinnamon cassia, ho wood essential oil O/W (8:2)	-	- the film with CNF-stabilized Ho wood showed excellent mechanical properties - films with CNF-stabilized essential oil showed higher thermal stability - films with CNF-stabilized essential oil showed a lower melting temperature and slightly increased water vapor permeability	[88]
Biomedical	Antibacterial activity of emulsions stabilized by CNC and CNF	0.1–1.0 wt.%	Cinnamaldehyde, eugenol, limonene O/W (1:9 to 4:6)	CNC (14–34µm) and CNF (27–51 nm) in droplet size	- CNC and CNF were able to stabilize as high as 40 wt.% oil - emulsions showed good stability towards storage and mild centrifugation - at low emulsion concentrations, CNF appeared to exhibit better antibacterial activity than CNC	[90]
	Encapsulation of coumarin and curcumin stabilized by aminated-CNC	0.1, 0.2, 0.3 wt.%	Natural coconut oil O/W (5:90 to 10:80)	≥150 nm in droplet size	- coumarin and curcumin were successfully encapsulated by aminated-CNC and maintained good stability - encapsulated Pickering emulsion exhibited excellent in vitro cytotoxicity for anticancer and antimicrobial effects - curcumin encapsulations showed a higher release profile than coumarin	[91]
	Encapsulation of vitamin D_3_ stabilized by CNF	0.1–0.7% (*w*/*w*)	Oil in water Soybean oil O/W (1:9) 9–24 µm in droplet size		- increasing the CNF concentration decreases the rate and extent of lipid digestion, and the vitamin bioaccessibility - by controlling the concentration of CNF, the extent and rate of lipid digestion can be determined for suitable use	[92]
	Thermal super-insulating material made of CNF-stabilized Pickering emulsions	0.2–30 g/L	Oil in water Hexadecane O/W (2:8) 10–20 µm in droplet size		- NFC-stabilized emulsions were able to be used as a pre-cursor for the formation of bio-aerogels - the bio-aerogel exhibited a very low thermal conductivity and high mechanical toughness	[93]
	Dual stimuli-responsive Pickering emulsion stabilized by Fe_3_O_4_ and CNC nanocomposite	0.05 wt.%	Oil in water Palm olein O/W (3:7) 11.90–109.00 µm in droplet size		- the prepared nanocomposite-stabilized Pickering emulsion was pH and magnetic responsive - the emulsion was stable at pH 3–6, then its stability was reduced from pH 8–10 and slightly improved at pH 12 - the attractability of the emulsion had a direct relation with the emulsion droplet diameter - the storage stability showed that gravitational separation does not affect the colloidal stability	[96]
Cosmetic	Poly (methyl methacrylate) latex stabilized by methyl cellulose-coated CNC	0.0021% (*w*/*v*)	Methyl methacrylate O/W (3:7)	15 ± 3 µm and ~100–200 nm in droplet size	- a higher CNC and methyl cellulose ratio formed more polymer microparticles while a lower ratio gave more nanoparticles - the latex morphology and roughness depend on the ratio of CNC and methyl cellulose and the later drying process	[94]
	Redispersible formulation of BC with carboxymethyl cellulose, CMC as oil-in-water emulsion stabilizer	0.1, 0.25, 0.50%	Isohexadecane O/W (1:9)	<10 µm in droplet size	- BC: CMC formulation was found to be a superior stabilizer than dry commercial celluloses - the formulation was able to stabilize the emulsion against coalescence and creaming for up to 90 days at a concentration of 0.50%	[97]
	Preparation of bio-based polymer by TEMPO-oxidized BC nanofibers for skincare applications	-	-	-	- the oil-in-water emulsion containing TEMPO-oxidized BC nanofibers prevented the carbon black from entering the microgrooves on the surface of the skin	[98]

**Table 4 molecules-27-07170-t004:** Summary of the scientific studies on the different surface wettability of CNC and CNF derivatives in their use as Pickering emulsifiers.

	Modification of NC	Change in Contact Angle (Unmodified → Modified)	Particle Content in Emulsion	Oil Phase Type, (Ratio of Oil:Water)	Emulsion Droplet Size (Oil Phase of Droplet)	Summary of Findings	Ref
CNC	Esterification with OSA	56°→ 80.2°	1 wt.%	Mixture of sunflower oil, tripropionin, and tributyrin (1:4)	1.22 µm	- resistance to coalescence and responsiveness to flocculation at a bio-relevant pH and ionic strength	[125]
	Esterification with OSA	51.7° → 82.1 and 85.0°	1.2 wt.%	Soy oil (8:2)	30–70 µm	- even at low concentrations, a stable and gel-like emulsion with fine droplets could be formed	[126]
	Modification with phenyl-trimethylammonium chloride	-	3–5 g L^−1^	Hexadecane (3:7)	2.4 µm	- a small amount of modified CNC showed a similar emulsifying ability to a traditional surfactant	[127]
	Grafting of polystyrene via reductive amination	-	1 wt.%	Toluene and hexadecane (1:2)	14 µm (Toluene) 4.8 µm (Hexadecane)	- modified CNC displayed superior surface properties and better stability emulsion than pristine CNC	[128]
CNF	Grafting of cinnamoyl chloride	46.10° → 51.43° and 68.36°	0.5 wt.%	Toluene and hexadecane (1:2)	≈32.5 µm for 51.43° and 5.6 µm for 68.36° (Toluene) ≈2.5 µm (Hexadecane)	- modified CNF showed favorable surface properties to stabilize toluene or hexadecane/water emulsion	[129]
	Adsorption of soy protein isolate	65° → 69°	0.08–0.24%	Canola oil (1:1)	10–40 μm	- modified CNF was able to form a uniform and stable emulsion and exhibited improved oxidative stability and anti-digestibility	[130]
	Different degrees of residual lignin in CNF	29° and 34°	0.75–2 mg mL^−1^	Dodecane (2:8)	≈16 µm	- CNF with a higher lignin content performed better as a Pickering stabilizer, but the emulsion can be finely tuned based on the residual lignin content in CNF	[131]
	Acetylation of CNF	30° and 81°	0.5%	Toluene (1:1)	10–60 µm for 81° and 20–80 µm for 30°	- CNF with a smaller size and higher contact angle produced small droplets while a larger size and lower contact angle produced larger droplets	[132]

**Table 5 molecules-27-07170-t005:** Summary of scientific studies on the different surface charges of CNC and CNF derivatives in their use as Pickering emulsifiers.

	Modification of NC	Zeta Potential (Unmodified → Modified)	Particle Content in Emulsion	Oil Phase Type, (Ratio of Oil:Water)	Emulsion Droplet Size	Summary of Findings	Ref
CNC	Different surface charge densities of sulphated CNC	−30, −43, −60 mV	≥4 mg mL^−1^	Hexadecane (3:7)	≈10–20 µm	- all 3 CNCs were able to stabilize hexadecane-in-water emulsion in the presence of salt	[2]
	Different sulfur content of sulphated CNC	−30.49, −37.35, −47.96 mV	0.3 wt.%	Medium-chain triglyceride oil (3:7)	18.01, 10.14, 2.97 µm	- the highest surface charged CNC displayed the best storage stability and surface coverage of emulsion droplets	[134]
	Desulfation of sulphated CNC by acid or basic desulfation	(−42.5 mV → −15, −25 mV)	0.5–20 mg mL_oil_^−1^	Dodecane (1:1)	≈10–500 µm	- lower surface charge of aggregated CNC (acid desulfation) formed smaller emulsion droplets but lower coverage ratio and vice versa	[135]
	Desulfation of sulphated CNC by mild acid treatment	-	≥10 mg mL^−1^	Hexadecane (1:9, 2:8, 3:7)	≈4 µm	- desulfation of CNC caused aggregation at the interfacial layer of emulsion while a monolayer of CNC was observed at the interface	[136]
CNF	TEMPO-oxidized CNF	-	3–4 mg mL^−1^	Hexadecane (2:8)	0.1–0.6 µm	- Using a high-pressure homogenizer, TEMPO-oxidized CNF was able to form nanoemulsions while unmodified CNF was unable to do so	[137]
	TEMPO-oxidized CNF with different degrees of oxidation	(−2.6 mV → −50.8, −59.4 mV)	0.1 wt.%	Palm fruit oil (1:10)	≈15 µm	- oxidized CNF showed improved emulsification efficiency and a suitable use in oil digestion inhibition application	[138]
	CNF with different surface charge densities by enzymatic treated (low charge) and TEMPO-oxidized (high charge)	-	0.5 wt.%	Rapeseed oil (2:3)	10–100 µm	- low charged CNF was more suitable to stabilize food emulsion where NaCl or acid are present, while highly charged CNF might be more suited for the application required for high emulsion viscosity at low concentrations	[139]
	Cationization of CNF by glycidyl trimethylammonium chloride	(−37 mV → +24 and +37 mV)	0.5 and 1 wt.%	Almond oil (3:7)	≈16–30 µm	- both cationized CNFs showed excellent emulsion stability when compared to their anionic analogue	[121]

**Table 6 molecules-27-07170-t006:** Summary of scientific studies on the different morphology of CNC and CNF derivatives in their use as Pickering emulsifiers.

	Modification of NC	Size	Particle Content in Emulsion	Oil Phase Type, (Ratio of Oil:Water)	Emulsion Droplet Size (Oil Phase of Droplet)	Summary of Findings	Ref
CNC	High-pressure homogenization as a post-treatment for CNC	Length 406–1500 nm Diameter 24.72–50.21 nm Aspect ratio 16.58–31.96	0.15%	Corn oil (1:1)	≈10–30 µm	- high-pressure homogenization could be applied as a post-treatment for CNC to alter their morphology, resulting in the formation of nanoparticles as effective Pickering emulsion stabilizers	[83]
	Various aspect ratios of CNC were obtained from different origins	Length 189, 855, ≈4000 nm Diameter 13, 17, 20 nm Aspect ratio 13, 47, 160	2 and 5 g L^−1^	Hexadecane (3:7)	4–10 µm	- all studied CNCs were able to form ultrastable emulsion under diluted conditions, but the different aspect ratios affected their coverage ratio of droplets	[120]
	Varying duration of acid hydrolysis	Length 178.2–261.8	1 wt.%	Palm oil (3:7)	1–10 µm	- smaller CNC had higher emulsification efficiency, but the addition of NaCl or casein affected the emulsion properties	[140]
CNF	High-pressure homogenization as a post-treatment for BCNF	Length >3 µm Diameter ≈30–230 nm	0.1–0.5 wt.%	Dodecane (1:9–5:5)	11–40 µm	- the concentration and physical size of CNF had a notable effect on the interface of the emulsion droplets	[84]
	Different duration of acid hydrolysis of modified CNF	Length >1 µm Diameter 295.3 and 575.8 nm	0.5 wt.%	Toluene and hexadecane (1:2)	≈5–20 µm (Toluene) ≈2.5 µm (Hexadecane)	- difference in size between the 2 studied CNFs affected the stabilizing mechanism and distribution of emulsion droplets	[129]
	TEMPO-oxidation of cellulase-treated CNF	Length 0.6–1.0 µm Diameter 10–20 nm	0.075–0.9 wt.%	Sunflower oil (2:8)	<10 µm	- TEMPO-CNF was a better Pickering stabilizer than the unmodified CNF as seen by the reduced emulsion droplet size	[141]

## Data Availability

Not applicable.

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
