# Peer review of "Review of Functional Aspects of Nanocellulose-Based Pickering Emulsifier for Non-Toxic Application and Its Colloid Stabilization Mechanism"

_molecules, 2022, doi:10.3390/molecules27217170_

Round 1
Reviewer 1 Report
It's a good review report about nanocellulose-based Pickering emulsifiers and could be published in Molecules. However, the authors here should not only summarize the reported works but also offer their understanding of the prospect of cellulose-based Pickering emulsifiers.
Reviewer 2 Report
Application of nanocellulose as Pickering emulsifier is written in an elaborate manner. Good work.
Now a days synthesis of nanocellulose is commonly demonstrated. But its applications are not explored more. The authors have written an elaborate review on the subject with more references.
Reviewer 3 Report
The manuscript presents an overview on the use of different types of nanocellulose (Cellulose nanocrystal (CNC), Cellulose nanofibril (CNF), and Bacteria cellulose (BC) as safe substitutes of inorganic particles as Pickering emulsifier.
The review adresses all the posible aspects, from the characteristics of different types of nanocellulose, the preparation and functionalization methods, applications in food, cosmetic, and biomedical application, the toxicity profile and the risk assessment guideline of whole Life Cycle. In addition, the findings on stabilization mecanism in Pickering emulsion and the criteria for effective emulsion stabilization are discused.
In my opinión, the review is very complete well written and structured and it can be published as it is.
Reviewer 4 Report
The particle-stabilized emulsion (Pickering emulsion) has become an important hot research. In addition to inorganic particles, nanocellulose and its derivatives are also used in the stabilization of liquid-oil interfaces. In this manuscript, the characteristics and applications nanocellulose-based Pickering emulsion, the mechanism of nanocellulose-stabilized liquid-oil interfaces, and the factors affecting the stabilization of Pickering emulsion are mainly reviewed. The research accords with the scope of Molecules - Speicial issue: Biomass-Derived Nanomaterials: Sustainable Production and Application. Overall, this manuscript is clearly organized, and the important points about nanocellulose-stabilized liquid-oil emulsion are presented. This review is very interesting for readers. After revision, this manuscript can be accepted for publishing.
Please note the following issues:
The review is too long, and seems to be a chapter of a book.
In the Introduction Section, 1.1 Lignocellulosic biomass and 1.2 Nanocellulose are too long (Pages 2-12), whereas total length is about 40 page. So it would be better if these sections are shortened.
Authors should focus on the review relative to nanocellulose-based Pickering emulsifier.
If possible, the applications of CNC- and CNF-based Pickering emulsion may be reviewed as a separated section.
If the mechanism of CNC or CNF-stabilized emulsion system is first described and then the factors affecting stabilization is discussed, it would be better.
Regarding 3. Factors affecting stabilization profile of CNC- and CNF-based pickering emulsion: The research results form different authors are discussed. They are important. However the proper classifications are not done. A better writing might be to classify them. Thus, readers are more interested in reading, and can obtain more information and help.
English writing should be carefully checked and improved.
For examples:
Line 44: “This has resulted of growing market”: “resulted of” should be replaced by “resulted in”.
Lines 49-50: “Thus, stabilization of surfaces and interfaces in emulsion is a key issue for wide range of product development.”: “stabilization” and “wide range of” should be replaced by “the stabilization” and “a wide range of”, respectively.
Line 52: “interfacial tension” should be replaced by “the interfacial tension”.
Line 54: “The interfacial of Pickering particles” should be replaced by “The interface of Pickering particles”.
Line 56: “that anchored at both interface” should be improved.
Lines 75-80: The descriptions should be improved.
Line 90: “The three main composition of lignocellulosic biomass are”: The grammar should be improved.
